# Synapse: Adaptive Arbitration of Complementary Expertise in Time Series Foundational Models

Sarkar Snigdha Sarathi Das[1,2,†], Palash Goyal[1], Mihir Parmar[1], Yiwen Song[1], Long T. Le[1],
Lesly Miculicich[1], Jinsung Yoon[1], Rui Zhang[2], Hamid Palangi[1,*], and Tomas Pfister[1,*]
[1]*Google,* [2]*Pennsylvania State University*
*Corresponding author(s): sfd5525@psu.edu, {palashgoyal,hamidpalangi}@google.com*
[†] *Work done while Sarkar was a Student Researcher at Google.*
[*]*Joint last authors*

**Reviewed on OpenReview:** *https://openreview.net/forum?id=j3HqbsCwt1*

## Abstract

Pre-trained Time Series Foundational Models (TSFMs) represent a significant advance, capable of forecasting diverse time series with complex characteristics, including varied seasonalities, trends, and long-range dependencies. Despite their primary goal of universal time series forecasting, their efficacy is far from uniform; divergent training protocols and data sources cause individual TSFMs to exhibit highly variable performance across different forecasting tasks, domains, and horizons. Leveraging this complementary expertise by arbitrating existing TSFM outputs presents a compelling strategy, yet this remains a largely unexplored area of research. In this paper, we conduct a thorough examination of how different TSFMs exhibit specialized performance profiles across various forecasting settings, and how we can effectively leverage this behavior in arbitration between different time series models. We specifically analyze how factors such as model selection and forecast horizon distribution can influence the efficacy of arbitration strategies. Based on this analysis, we propose SYNAPSE, a novel arbitration framework for TSFMs. SYNAPSE is designed to dynamically leverage a pool of TSFMs, assign and adjust predictive weights based on their relative, context-dependent performance, and construct a robust forecast distribution by adaptively sampling from the output quantiles of constituent models. Experimental results demonstrate that SYNAPSE consistently outperforms other popular ensembling techniques as well as individual TSFMs, demonstrating SYNAPSE's efficacy in time series forecasting.

## 1 Introduction

Time series forecasting is a pivotal task that underpins decision-making in high-stakes domains, from managing energy grids (Zhou et al., 2021; Lai et al., 2018), and supply chains (Mancuso et al., 2021) to navigating financial markets (Godahewa et al., 2021). Historically, the diversity of time series patterns necessitated a specialized approach, compelling practitioners to select different algorithms for different types of time series data. The advent of deep learning, particularly large-scale Transformer-based architectures, have sought to create a universal forecasting solution, pushing the boundaries of state-of-the-art performance for this task by learning time series foundation models (Das et al., 2024; Graf et al., 2025; Woo et al., 2024; Cohen et al., 2025; Shi et al., 2024). These models are trained on large corpora of time series data to learn the ability to identify specific patterns in numerical data. At convergence, the model weights capture the average dynamics of the training distribution.

However, this monolithic approach harbors a fundamental flaw: it forces the model into a ***representational compromise***. Real-world time series are inherently non-stationary, composed of a complex mixture of competing patterns-stable seasonalities, long-term trends, and abrupt structural breaks. A single, static model averages these conflicting signals, rendering it brittle and unprepared for sudden regime shifts, where

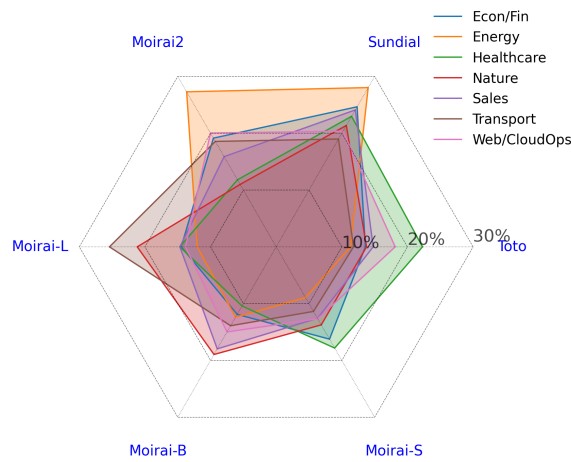

| Domain | Horizon | Modal (Oracle) | Entropy (Bits) | Modal Share | Switch Freq. % |
|--------|---------|----------------|----------------|-------------|----------------|
| **Econ/Fin** | Short | Sundial | 2.541 | 0.251 | 14.47% |
| **Energy** | Long | Moirai2 | 2.432 | 0.281 | 45.31% |
| | Medium | Sundial | 2.421 | 0.288 | 44.86% |
| | Short | Sundial | 2.467 | 0.278 | 43.53% |
| **Healthcare** | Short | Sundial | 2.531 | 0.237 | 41.28% |
| **Nature** | Long | Sundial | 2.505 | 0.283 | 29.36% |
| | Medium | Sundial | 2.497 | 0.285 | 27.79% |
| | Short | Moirai-L | 2.544 | 0.213 | 40.82% |
| **Sales** | Short | Sundial | 2.550 | 0.242 | 47.27% |
| **Transport** | Long | Moirai-L | 2.494 | 0.278 | 51.75% |
| | Medium | Moirai-L | 2.529 | 0.254 | 51.08% |
| | Short | Sundial | 2.550 | 0.236 | 58.45% |
| **Web/ CloudOps** | Long | Moirai2 | 2.562 | 0.225 | 49.31% |
| | Medium | Moirai2 | 2.556 | 0.223 | 50.09% |
| | Short | Toto | 2.509 | 0.240 | 49.67% |

Figure 1: (Left) Oracle arbitrator's model selection frequency across seven different domains of GIFT-Eval (Aksu et al., 2024). These values represent the percentage of timestamps each model was selected by Oracle as the optimal predictor across different domains. (Right) Detailed Oracle analysis by domain and forecast horizon. The **Modal (Oracle)** column identifies the Time Series Foundation Model (TSFM) most frequently selected as the optimal predictor by the Oracle, while **Modal Share** quantifies the proportion of time that model held that majority. **Selection Entropy** (where 2.58 bits denotes a uniform distribution across 6 models) provides a quantitative measure of selection diversity, and **Switch Frequency** represents the average percentage of timesteps where the Oracle's preferred model changes. These results demonstrate that model expertise is broadly distributed across the pool; the high entropy values and low modal shares provide a strong empirical justification for our arbitration approach, as a static selection would be consistently suboptimal.

performance can degrade catastrophically. While simple static ensembles are common mitigation strategy, they merely average these already-compromised representations. While this need for dynamic specialization is conceptually related to Mixture of Experts (MoE) frameworks (Shazeer et al., 2017; Zhou et al., 2022; Liu et al., 2024), where a gating network routes inputs to specialized sub-models, MoE architectures are typically trained end-to-end, jointly optimizing the gate and their constituent experts. Huang et al. (2025) explores framing this problem as multi-specialist cooperation by decomposing forecasting into sub-tasks handled by specialized agents. Our work addresses a different and, in the context of TSFMs, more pressing challenge: performing post-hoc arbitration between *pre-existing, monolithic, and fixed-weight* foundational models.

To validate this paradigm, we first quantify its empirical upper bound using a temporal Oracle that, at each timestamp, selects the most accurate prediction from a diverse pool of expert models. Examining the selection frequency of optimal model prediction at timestamp level in Figure 1 (left), we observe that optimal model can vary widely across different timestamps, and their usage percentages also vary widely across different domains. We find that the Oracle frequently relies on weaker, specialized models that excel only in specific regions. Furthermore, the number of model switches required for optimal prediction is substantial (Figure 1, right), proving static selection is insufficient. This evidence highlights the fundamental constraints of traditional ensembles. Such approaches apply a fixed aggregation rule-like centering predictions or taking the median of quantiles (Garza & Rosillo, 2025), which overlooks informative output distributions, yields suboptimal performance (Table 1), and is fundamentally unable to adapt to the highly dynamic, timestamp-to-timestamp nature of model performance.

Informed by these observations, we introduce SYNAPSE, a novel TSFM arbitration framework for time series forecasting. SYNAPSE keeps a pool of TSFMs, and dynamically arbitrates between them at timestamp level granularity to leverage the predictions from the best forecasting models, while de-prioritizing the less accurate models for a specific timestamp. For each timestamp, SYNAPSE merges the inverse quantile predictions from its constituent models to give the final output distribution. To dynamically adjust the arbitration at

timestamp level, Synapse maintains a moving window, which it leverages for selecting and adjusting the weights for arbitration.

Our experimental results, conducted on the extensive GIFT-eval benchmark of 23 datasets, provide strong evidence for the superiority of the dynamic arbitration paradigm. Synapse achieves new state-of-the-art performance, with performance gains consistently amplifying over longer forecast horizons. The strength of Synapse is highlighted by its ability to consistently arbitrate a committee of specialists to outperform any single member, demonstrating its potential to sometimes surpass a single, monolithic stronger model. This result proves that the adaptive arbitration mechanism itself, is the primary driver of performance. Our primary contributions are:

- We identify the "representational compromise" inherent in monolithically trained models as a core weakness for non-stationary time series forecasting. Subsequently, we built an "Oracle" arbitrator, that optimally selects the best model forecast at each timestamp to set an empirical performance upper bound in time series forecasting with current models.

- We re-frame the forecasting task and define a dynamic arbitration problem between TSFMs. We then propose Synapse, a novel arbitration framework that dynamically arbitrates the influence of constituent models at each time step.

- We show that Synapse achieves state-of-the-art performance, with gains that amplify over longer horizons, where baseline ensemble approaches falls apart due to static mixture of ensembles. We directly link this to its dynamic weight-shifting mechanism.

- Finally, we demonstrate that the arbitration mechanism is the key driver of performance, enabling Synapse to effectively synthesize complementary expertise and consistently improve upon the capabilities of its individual constituent models.

Our work establishes that adaptive arbitration is a powerful and necessary paradigm for building the next generation of robust, real-world time series forecasting systems.

## 2 Related Works

The field of time series forecasting has evolved through several distinct paradigms, starting with statistical methods, advancing to task-specific deep learning models, and most recently leveraging large-scale foundational models. Our work is situated at the intersection of this latest paradigm and the long-standing practice of model ensembling, adapted for the unique challenges and opportunities presented by modern TSFMs.

**Statistical and Deep Learning Methods** Time series analysis has historically relied on statistical methods like the ARIMA family (Box et al., 2015; Box & Jenkins, 1976), ETS (Hyndman et al., 2008), and additive regression (Taylor & Letham, 2018). While robust baselines, they often struggle with complex non-linear dependencies. The advent of deep learning introduced automated pattern learning, evolving from RNNs (Hochreiter & Schmidhuber, 1997) and probabilistic models (Salinas et al., 2020) to specialized architectures like N-BEATS (Oreshkin et al., 2019) and TimesNet (Wu et al., 2023). Recently, the field has debated the efficacy of Transformers (Zhou et al., 2021; Wu et al., 2021) versus simpler MLP-based models (Chen et al., 2023; Zeng et al., 2023), with innovations like PatchTST (Nie et al., 2023) bridging the gap. Despite their power, these deep learning models are typically trained for specific tasks and datasets, requiring significant data and computational resources for each new application.

**Time Series Foundational Models** The latest paradigm shift in forecasting mirrors the revolution in natural language processing with large language models: the rise of pre-trained Time Series Foundational Models (TSFMs). These models, trained on massive and diverse datasets, perform zero-shot forecasting on unseen time series with remarkable accuracy. A prominent approach treats forecasting as a language modeling problem, where continuous time series values are tokenized into a discrete vocabulary. Models like Chronos (Ansari et al., 2024) and TimeGPT (Garza et al., 2023) exemplify this, using standard large language

model (LLM) architectures to predict the next token. Building on this, Chronos-2 (Ansari et al., 2025) evolves the architecture from a univariate predictor to a "universal" forecaster, leveraging a group attention mechanism and synthetic training data to natively handle multivariate dependencies and external covariates. A contrasting approach involves direct regression on continuous values. TimesFM (Das et al., 2024) is a leading example, employing a decoder-only architecture pre-trained on 100 billion time points for extreme long-horizon forecasting. Other models explore novel pre-training strategies and architectures, including generative state-space models like Flowstate (Graf et al., 2025). The Moirai family (Woo et al., 2024) particularly illustrates rapid architectural evolution: starting as a masked encoder with multi-patch layers to handle diverse frequencies, it evolved into Moirai-MoE (Liu et al., 2024) utilizing a sparse Mixture-of-Experts. Most recently, Moirai 2.0 (Liu et al., 2025a) shifts towards a more streamlined decoder-only architecture, utilizing quantile loss to achieve superior probabilistic accuracy and inference speed with significantly fewer parameters.. The core promise of TSFMs is a universal forecasting solution, yet as our Oracle analysis demonstrates, no single model universally dominates. Instead, they exhibit complementary expertise, making them prime candidates for advanced arbitration.

## 3 Preliminaries and Insights

### 3.1 Time Series Forecasting

The fundamental task of time series forecasting is to predict future values of a sequence based on its observed history. Therefore, this problem can be framed as a mapping from historical context data to future horizon prediction.

Formally, let $\mathbf{X}_{1:\tau} = (x_1, x_2, \ldots, x_\tau)$ represent a univariate/multivariate time series of $c$ variables observed over a context window of length $L$. The primary objective is to generate a forecast for the subsequent $T$ timesteps, known as the forecast horizon, denoted $\mathbf{X}_{\tau+1:\tau+T}$. Modern forecasting approaches move beyond simple point estimates to generate a full predictive distribution, $P(\mathbf{X}_{\tau+1:\tau+T}|\mathbf{X}_{1:\tau})$, which captures the inherent uncertainty through forecast quantiles.

The quality of this forecast is then evaluated against the ground truth, $\mathbf{Y} = \mathbf{X}_{\tau+1:\tau+T}$, using metrics that assess both probabilistic accuracy and point-wise error. Commonly used metrics for these are Continuous Ranked Probability Score (CRPS) and Mean Absolute Scaled Error (MASE) respectively. Details about these metrics can be found in Appendix B.

### 3.2 Arbitration of Time Series Forecasting Models - Problem Formulation

Let $\mathcal{M} = \{M_1, M_2, \ldots, M_N\}$ be a pool of $N$ distinct Time Series Foundational Models (TSFMs). For a given input $\mathbf{X}_{1:\tau}$, each model $M_i \in \mathcal{M}$ generates its own predictive distribution $P_i(\mathbf{X}_{\tau+1:\tau+T}|\mathbf{X}_{1:\tau})$ for the forecast horizon.

Our proposed arbitration setting aims to synthesize these individual forecasts into a single, superior predictive output. Instead of a simple aggregation, we define a dynamic arbitration process where the final forecast is a predictive distribution, constructed by an arbitration function, $\mathcal{A}$, that operates on constituent models' output distributions.

The arbitrated predictive distribution for timestep $t$, represented by its set of quantiles $\hat{Q}_t^{\mathrm{arb}}$, is given by:

$$\hat{Q}_t^{\mathrm{arb}} = \mathcal{A}(\{(w_{i,t}, \hat{Q}_{i,t}) \mid M_i \in \mathcal{M}\})$$

where:

- $\hat{Q}_{i,t}$ represents the predictive quantile distribution from model $M_i$ for timestep $t$.

- $w_{i,t}$ is a dynamic, non-negative weight assigned to model $M_i$ at timestep $t$, reflecting its expected performance.

This framework can be specialized to represent traditional ensembles. For example, a simple mean ensemble of point forecasts can be formulated when weights are uniform ($w_{i,t} = 1/N$), and $\mathcal{A}$ is defined as the averaging function over the median of each input distribution $\hat{Q}_{i,t}$.

For the scope of this work, we limit our experiments to a pool of six highly popular Time Series Foundational Models: Sundial (Liu et al., 2025b), Toto (Cohen et al., 2024), Moirai2, Moirai-large, Moirai-base, and Moirai-small. (Woo et al., 2024).

### 3.3 Oracle Selection Arbitrator

To establish an empirical performance expectation for a selective arbitration strategy, we define an *Oracle-Based Selection Arbitrator*. This conceptual model operates with perfect foresight, possessing knowledge of the ground truth outcomes. At each timestep $t$ in the forecast horizon, the Oracle selects the single, optimal TSFM from the pool $\mathcal{M}$, that is, the model whose predictive distribution is closest to the actual value.

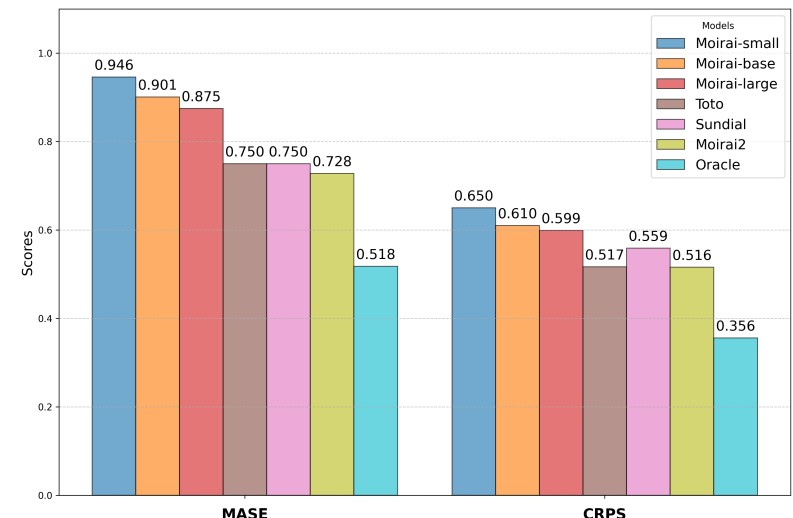

Figure 2: GIFT-Eval Performance comparison (MASE | CRPS) of the Oracle Arbitrator against individual constituent TSFMs. Oracle selector can outperform all other constituent models by a large margin, demonstrating the efficacy of a strong arbitrator.

Formally, the predictive distribution selected by the Oracle at timestep $t$ is $\hat{Q}_t^{\text{oracle}}$, defined as:

$$\hat{Q}_t^{\text{oracle}} = \hat{Q}_{i^*,t} \quad \text{where} \quad i^* = \arg\min_{i \in \{1,\dots,N\}} \text{CRPS}(\hat{Q}_{i,t}, y_{\tau+t})$$

Here, $y_{\tau+t}$ is the ground truth value at the future timestep.

The performance of this Oracle thus represents the lowest possible error achievable if the best model could be identified at every single step. The significant potential of dynamic arbitration is highlighted in Figure 2, which compares the performance of the Oracle against the individual TSFMs in our pool.

Furthermore, analyzing the Oracle's selection percentages reveals an important insight: no single model dominates across all timestamps. As illustrated in Figure 1 (left), the selection frequency of each TSFM varies considerably across different domains. Notably, models that exhibit weaker overall performance (e.g., Moirai-small) are still frequently chosen as the optimal predictor for specific timestamps, highlighting the context-dependent nature of forecasting expertise and reinforcing the need for an adaptive arbitration mechanism.

**Oracle Analysis and Selection Diversity** Beyond establishing an upper bound, the Oracle provides deep insights into the transient nature of model expertise. In Figure 1, we analyze the Oracle's choices across GIFT-Eval domains and horizons using four key metrics. If we let $p_i$ represent the empirical selection probability of model $M_i$ over the forecast horizon. The **Modal (Oracle)** choice identifies the model that holds the lead most frequently for being chosen ($i^* = \arg\max_i p_i$), while the **Modal Share** ($p_{i^*}$) quantifies the percentage of timesteps the most frequent model held the lead. To measure selection diversity across the pool, we compute the **Selection Entropy**:

$$H = -\sum_{i=1}^{N} p_i \log_2 p_i \tag{1}$$

In our experiments, $N = 6$ for the pool of six TSFMs, therefore the maximum theoretical entropy for this pool is $\log_2(6) \approx 2.585$ bits, representing a perfectly uniform distribution of expertise.

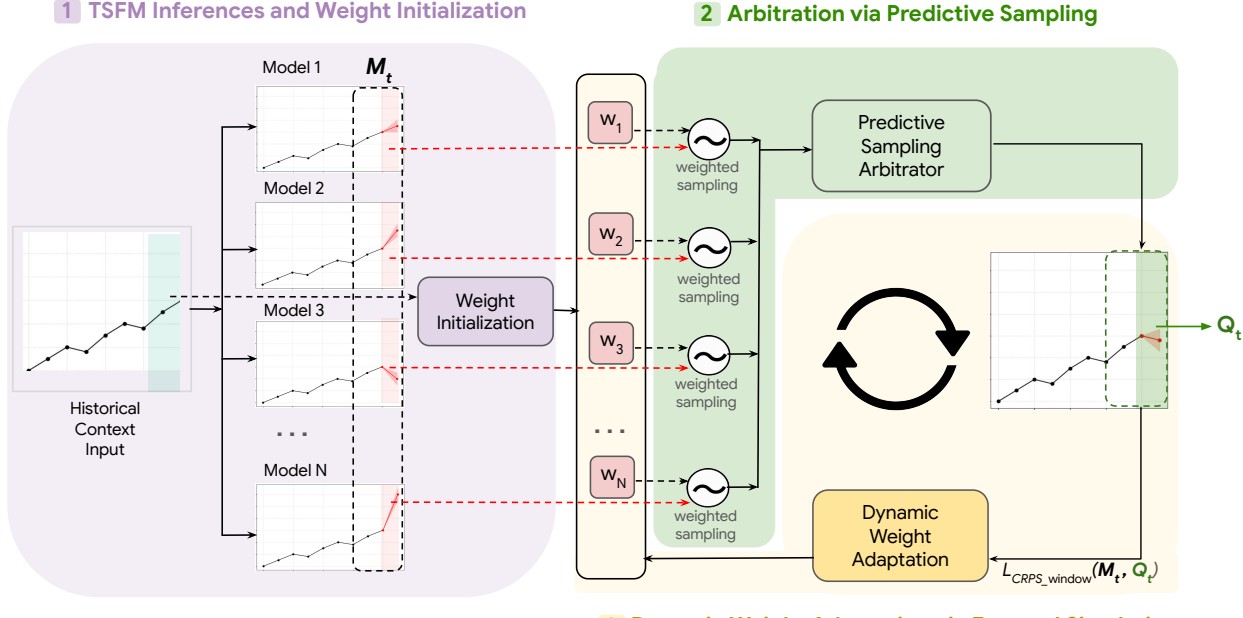

Figure 3: **Overview of Synapse.** It takes a *Historical Context Input* (left), which is then fed into a pool of $N$ diverse *Time Series Foundational Models (TSFMs)*. Each model produces its probabilistic forecast. These individual forecasts are then fed into the core arbitration mechanism. At each timestep $t$, a set of *Dynamic Weights* $\{w_1, \ldots, w_N\}$ is applied to the corresponding model forecasts, which is leveraged in the predictive sampling and in subsequent construction of the final *Arbitrated Forecast Distribution* for that timestep. This entire process is governed by a *Dynamic Weight Adaptation* via Forward Simulation loop: the arbitrator's own forecast from previous step is used as a simulated observation, which is then used to update a rolling performance window, which recalculates the CRPS for all models to determine the dynamic weights $\{w_{i,t+1}\}$ for the next timestep.

As shown in the table in Figure 1 (right), the **Selection Entropy** values consistently exceed 2.4 bits, approaching the theoretical maximum. This demonstrates that Oracle optimality is broadly distributed across the pool rather than being concentrated on a single dominant model with rare "flips." Furthermore, the **Modal Share** remains low (typically 20%-28%), meaning that even the best model in a domain is suboptimal for over 70% of the horizon. Finally, the high **Switch Frequency** (often $> 50\%$) confirms that model leadership is highly transient across timesteps. This persistent and high-frequency switching underscores that no single model is optimal for an entire forecast window. In fact, as we demonstrate in Appendix C, even the best domain-specific static model substantially underperforms a dynamic arbitration strategy. Together, these metrics demonstrate a fundamental "representational compromise" in monolithic TSFMs and justify the necessity of the dynamic arbitration provided by SYNAPSE.

## 4 Synapse: Dynamic Arbitration

Built on the definition of general arbitration framework (Section 3.2), SYNAPSE is designed to close the performance gap to the Oracle selector by dynamically adjusting its strategy at each step of the forecast horizon. As demonstrated in Figure 3, SYNAPSE's core mechanism consists of two key components: (1) a weighted arbitration function based on predictive sampling, (2) a dynamic weighting scheme based on a rolling performance window that performs a forward simulation mechanism allowing it to adapt over the forecast horizon. For this implementation, we utilize the full set of models at each step.

## 4.1 Arbitration via Predictive Sampling

We design the arbitration function, $\mathcal{A}$, to create a non-parametric mixture of the underlying distributions they represent. We achieve it through a weighted predictive sampling process, which guarantees that the resulting arbitrated quantiles, $\hat{Q}_t^{\mathrm{arb}}$, are probabilistically valid (i.e., monotonic) by construction, eliminating the need for post-processing.

**Sample Generation** For each model $M_i$, we generate $n_i$ random samples from its predictive distribution. In this regard, we apply inverse transform sampling method to the model's estimated quantile function. The final arbitrated quantile for a given level $\alpha$, denoted $\hat{q}_{\alpha,t}^{\mathrm{arb}}$, is then calculated as the empirical $\alpha$-th quantile of the pooled set of all generated samples. We formally express this entire process as:

$$\hat{q}_{\alpha,t}^{\mathrm{arb}} = \mathrm{Quantile}\left(\bigcup_{i=1}^{N}\left\{\hat{F}_{i,t}^{-1}(p_j) \mid p_j \sim U(0,1)\right\}_{j=1}^{n_i}, \alpha\right) \quad \text{where } n_i = \lfloor N_{\mathrm{total}} \cdot w_{i,t}\rceil \tag{2}$$

Here $n_i$ is the number of samples allocated to model $M_i$, determined by its performance weight $w_{i,t}$. Here, $\hat{F}_{i,t}^{-1}(\cdot)$ is the inverse Cumulative Distribution Function (CDF), estimated from the discrete quantile predictions of model $M_i$ at time $t$. The continuous function $\hat{F}_{i,t}^{-1}$ is practically approximated by fitting a cubic spline to the discrete quantile points for probabilities within $[0.1, 0.9]$, while stable linear extrapolation is used for the tails of the distribution.

This sampling process yields a pooled set of samples, $\mathcal{P}_t = \bigcup_{i=1}^{N}\{\text{samples from } M_i\}$. The final arbitrated predictive distribution, $\hat{Q}_t^{\mathrm{arb}}$, is the set of empirical quantiles calculated from this collection $\mathcal{P}_t$. This method ensures that the final distribution faithfully represents a true mixture of the constituent predictive distributions.

## 4.2 Dynamic Weight Adaptation via Forward Simulation

Although Equation 2 presents a compelling strategy of dynamically adjusting the weight of model predictions at different timesteps to account for the changing performance trends of different models, the primary challenge here is to calculate the weight in the absence of ground truth value for the prediction horizon. To address this, we employ a **forward simulation** to adapt the model weights, $\{w_{i,t}\}$, at each step. The weights are determined by each model's recent performance, as measured by its CRPS score over a rolling historical window of length $W$. The primary weighting rule is inverse error weighting ($w_{i,t} \propto 1/\mathrm{CRPS}_{i,t-W:t-1}$), with a robust softmax fallback to handle the numerical instability of near-zero scores. This entire adaptation mechanism operates as a feedback loop:

1. **Step $t$ Prediction:** At timestep $t$ in the horizon, the arbitrator calculates the final arbitrated quantiles, $\hat{Q}_t^{\mathrm{arb}}$, using the current set of weights $\{w_{i,t}\}$.

2. **Simulated Ground Truth:** It then defines a **simulated ground truth**, $y_t^{\mathrm{sim}}$, for that timestep by taking the median of its own prediction: $y_t^{\mathrm{sim}} = \mathrm{median}(\hat{Q}_t^{\mathrm{arb}})$

3. **Performance Window Update:** This simulated observation is used to update the rolling performance window. The oldest data point in the window is discarded, and a new record, consisting of the simulated ground truth ($y_t^{\mathrm{sim}}$) and the original predictions from all models for that step ($\{\hat{Q}_{i,t}\}$), is added.

4. **Weight Update:** The CRPS scores and, consequently, the weights $\{w_{i,t+1}\}$ for the *next* timestep, $t+1$, are then recalculated based on this newly updated performance window.

This process repeats for every step in the forecast horizon, allowing the arbitrator to dynamically learn and adapt its strategy based on a simulation of its own ongoing performance. Intuitively, this allows SYNAPSE to create a self-reinforcing feedback loop: models whose predictions align well with the aggregated mixture median (the simulated ground truth) will achieve a lower CRPS score in the subsequent step and thus receive

---

**Algorithm 1** SYNAPSE

---

1: **Input:** Pool of TSFMs $\mathcal{M} = \{M_1, \ldots, M_N\}$, Forecast horizon $T$, Initial performance window $\mathcal{W}_{\text{init}}$, Total samples $N_{\text{total}}$, Quantile levels $\{\alpha_k\}$
2: **Output:** Final arbitrated forecast distributions $\{\hat{Q}_t^{\text{arb}}\}_{t=1}^T$

3: $\mathcal{W} \leftarrow \mathcal{W}_{\text{init}}$          ▷ Initialize rolling performance window
4: $\{\hat{Q}_t^{\text{arb}}\}_{t=1}^T \leftarrow \{\}$          ▷ Initialize output container

5: **for** $t \leftarrow 1$ to $T$ **do**
6:     *// Dynamic Weight Calculation*
7:     **for** $i \leftarrow 1$ to $N$ **do**
8:         $s_i \leftarrow \text{AverageCRPS}(M_i, \mathcal{W})$          ▷ Calculate score for each model
9:     **if** $\min(\{s_i\}) \approx 0$ **then**
10:         $\{w_{i,t}\} \leftarrow \text{Normalize}(\text{Softmax}(\{-\gamma * s_i\}))$          ▷ fallback
11:     **else**
12:         $\{w_{i,t}\} \leftarrow \text{Normalize}(\text{InverseError}(\{s_i\}))$

13:     *// Arbitration via Predictive Sampling*
14:     $\mathcal{P}_t \leftarrow \emptyset$
15:     **for** $i \leftarrow 1$ to $N$ **do**
16:         $n_i \leftarrow \lfloor N_{\text{total}} \cdot w_{i,t} \rfloor$          ▷ Calculate number of samples to take from each model
17:         $\hat{F}_{i,t}^{-1} \leftarrow \text{FitSpline}(\hat{Q}_{i,t})$
18:         $\mathcal{P}_{i,t} \leftarrow \{\hat{F}_{i,t}^{-1}(p_j) \mid p_j \sim U(0,1)\}_{j=1}^{n_i}$
19:         $\mathcal{P}_t \leftarrow \mathcal{P}_t \cup \mathcal{P}_{i,t}$
20:     $\hat{Q}_t^{\text{arb}} \leftarrow \text{Quantile}(\mathcal{P}_t, \{\alpha_k\})$

21:     *// Forward Simulation and Window Update*
22:     $y_t^{\text{sim}} \leftarrow \text{median}(\hat{Q}_t^{\text{arb}})$
23:     $o_{\text{new}} \leftarrow (y_t^{\text{sim}}, \{\hat{Q}_{i,t}\}_{i=1}^N)$
24:     $\mathcal{W} \leftarrow (\mathcal{W} \setminus \{o_{\text{oldest}}\}) \cup \{o_{\text{new}}\}$          ▷ Update rolling window

25: **return** $\{\hat{Q}_t^{\text{arb}}\}_{t=1}^T$

---

a higher weight. On the other hand, models that consistently diverge from the emerging consensus are penalized with lower weights, effectively reducing their influence over time. While this forward simulation relies on the aggregated consensus as a proxy for ground truth, our ablation study (detailed in Table 3) demonstrates this is a key driver of SYNAPSE's success, as the SYNAPSE with dynamicity substantially outperforms static weights.

## 5 Experiments

In this section, we empirically validate the effectiveness of SYNAPSE. Our experimental setup is grounded in the framework established in our preliminaries, focusing on a pool of highly popular TSFMs. We conduct a series of experiments to evaluate SYNAPSE's overall performance, compare it against strong ensemble baselines. Furthermore, we perform a rigorous ablation study to dissect the components contributing to the improvements, followed by a performance analysis across different scenarios.

### 5.1 Experiment Setup

**Benchmark Dataset**    To demonstrate the effectiveness of SYNAPSE, we evaluate and compare its performance in GIFT-Eval comprehensive benchmark Aksu et al. (2024). This benchmark consists of 23 datasets

Table 1: Overall performance comparison on the GIFT-Eval benchmark, broken down by forecast horizon. The best model performance in each column is in **bold**, and the second best is underlined.

| Model | Long Horizon | | Medium Horizon | | Short Horizon | | Overall | |
|---|---|---|---|---|---|---|---|---|
| | CRPS | MASE | CRPS | MASE | CRPS | MASE | CRPS | MASE |
| Moirai2 | 0.513 | 0.789 | 0.519 | 0.759 | **0.517** | **0.695** | 0.516 | 0.728 |
| Toto | 0.496 | 0.812 | 0.499 | 0.772 | 0.533 | 0.720 | 0.517 | 0.750 |
| Sundial | 0.510 | 0.785 | 0.526 | 0.764 | 0.592 | 0.732 | 0.559 | 0.750 |
| Moirai-small | 0.626 | 1.035 | 0.636 | 1.021 | 0.666 | 0.888 | 0.650 | 0.946 |
| Moirai-base | 0.611 | 1.029 | 0.632 | 1.017 | 0.601 | 0.818 | 0.610 | 0.901 |
| Moirai-large | 0.598 | 0.985 | 0.605 | 0.961 | 0.597 | 0.807 | 0.599 | 0.875 |
| Mean Ensemble | 0.502 | 0.846 | 0.514 | 0.82 | 0.535 | 0.727 | 0.523 | 0.771 |
| Median Ensemble | 0.481 | 0.793 | 0.493 | 0.767 | 0.523 | 0.707 | 0.517 | 0.762 |
| **Synapse** | **0.464** | **0.756** | **0.472** | **0.733** | **0.517** | 0.7 | **0.496** | **0.719** |
| Oracle | 0.329 | 0.530 | 0.328 | 0.512 | 0.379 | 0.517 | 0.356 | 0.518 |

across 7 different domains having over 144,000 time series, featuring 97 unique combinations of dataset, frequency, and length. Table 1 shows the overall performance comparison of different methods in GIFT-Eval.

**Baselines** As denoted in Section 3.2, we primarily consider Sundial (Liu et al., 2025b), Toto (Cohen et al., 2024), Moirai2, Moirai-large, Moirai-base, and Moirai-small (Woo et al., 2024) as the foundational models. We compare SYNAPSE performance to that of these constituent models; additionally we compare against a recent strong baseline - median-ensemble (Garza & Rosillo, 2025) that takes median values of multiple models at each quantile, its variant mean-ensemble, and standard point forecast means as our baselines.

## 5.2 Overall Performance

Table 1 presents the CRPS and MASE scores for each of the six constituent TSFMs, median ensemble, and mean ensemble compared against the performance of SYNAPSE. The results clearly demonstrate that SYNAPSE significantly outperforms every individual model in the pool, as well as the ensemble baselines. Notably, it achieves a CRPS of 0.496, which is a marked improvement over the best-performing individual model, Moirai2 (0.516). This showcases the ability of our dynamic arbitration mechanism to successfully synthesize the varied expertise of the foundational models into a single, more accurate predictive distribution.

## 5.3 Comparison with Ensemble Baselines

For comparison with ensemble techniques, we choose median-ensemble (Garza & Rosillo, 2025), a recent strong ensemble baseline that leverages medians over quantiles for time series prediction. While the Median Ensemble (MASE 0.762) offers a slight improvement over some individual models, it fails to match the performance of the best-performing constituent TSFM (Moirai2, MASE 0.728). In contrast, SYNAPSE (MASE 0.719) substantially outperforms both the Median Ensemble and the best constituent individual model, confirming that its dynamic and adaptive nature provides a distinct advantage over simple aggregation strategies.

## 5.4 Performance Across Forecast Horizons

Table 1 also details the performance trends across different prediction horizons. To visualize these trends more clearly, we examine the line plots in Figure 4. The plots illustrate that as we move from short horizon to medium and longer horizon, the efficacy of SYNAPSE arbitration framework also grows. More specifically, for short-horizon forecasts, SYNAPSE's performance is competitive with the best-performing individual model for that range, Moirai2. However, as the task difficulty increases for medium and long-horizon predictions, SYNAPSE establishes a distinct performance advantage. It begins to more effectively

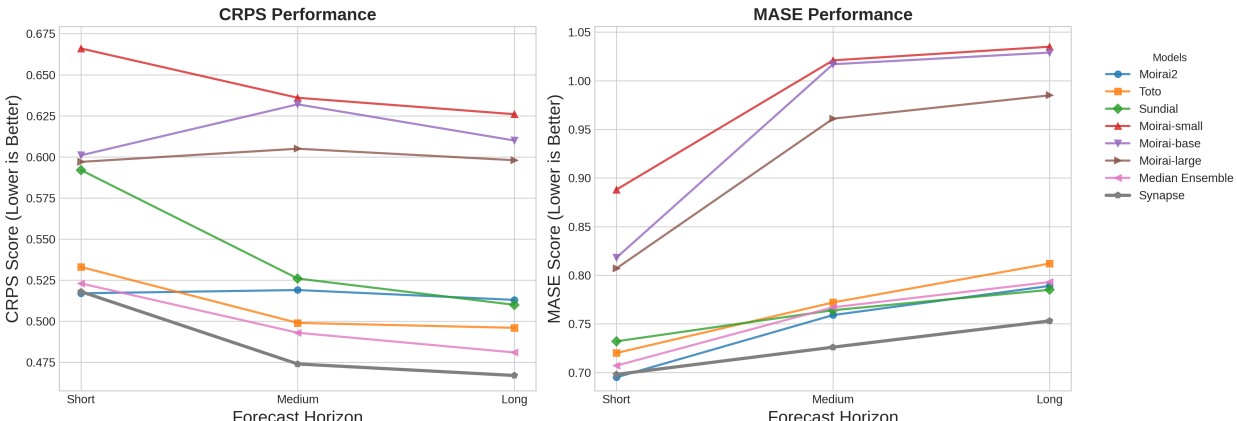

Figure 4: Performance of all models across different forecast horizons (Short, Medium, Long). SYNAPSE consistently shows strong and often superior performance for both CRPS and MASE, particularly as the forecast horizon increases.

leverage the complementary strengths of different models in the pool, widening the performance gap between itself and any single TSFM. In contrast, the Median Ensemble struggles to adapt, often exhibiting a higher MASE score than several of its constituent models at longer horizons, which indicates a failure to effectively aggregate their predictive power.

## 5.5 Model Selection Accuracy Comparison

While the ultimate goal of an ensemble is to produce the most accurate final forecast, it is also insightful to evaluate the effectiveness of the underlying arbitration mechanism itself. We define **Model Selection Accuracy** as a metric that quantifies how well the weighting of SYNAPSE aligns with the choices made by the *Oracle Arbitrator* (defined in Section 3.2) at each timestep. This metric assesses whether the arbitration correctly identifies the truly best-performing model (or ranks it highly) based on the information available to it. For comparison against primary baseline median-ensemble (Garza & Rosillo, 2025), since it does not do explicit weighting, we infer its "implicit choice" by ranking the constituent models based on the proximity of their median forecast to the final ensemble median forecast. Models whose median predictions are closer to the ensemble's final output median are given a better rank.

Table 2: Top-$k$ Model Selection Accuracy Comparison of SYNAPSE and Quantile Median Ensemble. Measures the percentage of timesteps where the Oracle's best model was ranked within the top $k$ ranking by the corresponding method.

| Method | Top-1 (%) | Top-2 (%) | Top-3 (%) | Top-4 (%) | Top-5 (%) | Top-6 (%) |
|---|---|---|---|---|---|---|
| Median Ensemble | 10.73 | 21.79 | 35.43 | 53.36 | 74.25 | 100 |
| SYNAPSE | **23.52** | **41.57** | **56.95** | **70.18** | **82.87** | **100** |

Table 2 presents the Top-$k$ Model Selection Accuracy for both the SYNAPSE arbitrator and the Median Ensemble baseline, averaged across all datasets and forecast horizons evaluated. The results demonstrate that the dynamic weighting mechanism employed by SYNAPSE is significantly more effective at identifying the best-performing model at each timestep compared to the implicit ranking of the Median Ensemble, effectively showing the efficacy of SYNAPSE in closing the gap with Oracle.

## 5.6 Impact of Arbitration and Dynamicity

To deconstruct the sources of SYNAPSE's performance gains, we conduct an ablation study to isolate the contributions of its two primary components: (1) Arbitration via Predictive Sampling and (2) Dynamic

Weight Adaptation. We compare three variants: (1) **Median Ensemble**, the baseline following Garza & Rosillo (2025) using no components from our proposed techniques; (2) **Synapse (- Dynamicity)**, which uses our predictive sampling arbitration but with static, uniform weights (removing dynamic adaptation); and (3) **Synapse**, the full proposed model.

The results in Table 3 clearly shows that shifting from a Median Ensemble to Synapse with static weights ('-Dynamicity') yields a notable performance improvement (MASE drops from 0.762 to 0.738), demonstrating the inherent value of the predictive sampling method for creating a superior mixture distribution. The subsequent addition of the dynamic weight adaptation via forward simulation provides another substantial boost, further reducing the MASE to 0.719. This two-step improvement validates the architectural choices of Synapse, proving that both the intelligent mixture strategy and the adaptive weighting scheme are crucial to its success.

Table 3: Domain-wise performance ablation for Synapse, detailing the incremental gains from its core components. We compare the baseline (Median Ensemble), Synapse without dynamic weighting, and the full Synapse. The best-performing model for each metric within each domain is highlighted in **bold**.

| Domain | Lumpiness | MASE | | | CRPS | | |
|---|---|---|---|---|---|---|---|
| | | Median Ensemble | Synapse (-Dynamicity) | Synapse (Full) | Median Ensemble | Synapse (-Dynamicity) | Synapse (Full) |
| Econ/Fin | 0.14 | 0.801 | 0.802 | **0.784** | 0.738 | 0.745 | **0.737** |
| Energy | 0.24 | 0.870 | 0.845 | **0.825** | 0.617 | 0.603 | **0.592** |
| Healthcare | 0.07 | 0.632 | 0.633 | **0.620** | 0.491 | 0.503 | **0.471** |
| Nature | 1.09 | 0.722 | **0.709** | 0.711 | 0.342 | **0.337** | 0.345 |
| Sales | 1.54 | **0.688** | 0.691 | 0.690 | **0.416** | 0.417 | **0.416** |
| Transport | 0.39 | 0.601 | 0.595 | **0.588** | 0.450 | 0.445 | **0.440** |
| Web/CloudOps | 1.47 | 0.808 | 0.737 | **0.690** | 0.561 | 0.534 | **0.497** |
| **Overall** | | 0.762 | 0.738 | **0.719** | 0.517 | 0.507 | **0.496** |

## 5.7 Domain Wise Performance Ablation

To further understand the behavior of our framework, we present a performance breakdown across the seven distinct domains within the GIFT-Eval benchmark. Table 3 details the performance of Synapse against Quantile median ensemble (Garza & Rosillo, 2025) in each domain. The results show that Synapse generally outperforms the static median-ensemble baseline. Please note that datasets in "Sales" domain (e.g. Restaurants, Hierarchical Sales, Car Parts) has substantially shorter forecast horizon (8-30) which does not provide sufficient runway for Synapse to meaningfully adapt the weights, Consequently, we see comparable results to that of Median Ensemble.

To further analyze this in the context of domain-specific data characteristics, conducting a correlation analysis between the performance difference (Median Ensemble MASE - Synapse MASE) and the lumpiness of each domain, we found a positive correlation score of *0.32* suggesting that the dynamic arbitration mechanism can be beneficial at navigating the inconsistent data patterns found in highly variable, non-stationary environments.

## 5.8 Synapse: Performance Scaling with Models

The primary strength of Synapse is its ability to synthesize the complementary expertise of multiple TSFMs. Therefore, it is imperative to understand how its performance scales as the pool of available models expands over time. We simulate this evolution by establishing a fixed, chronological order for the constituent models-Moirai-small, Moirai-base, Moirai-large, Sundial, Toto, and Moirai-2 and then constructing ensembles of incrementally increasing size. Starting with the first two models, we cumulatively add the next model in the sequence to form ensembles of three, four, five, and ultimately all six models.

At each step, we compare the performance of Synapse against the single best-performing model within that specific pool, based on its CRPS. The results, as shown in the Figure 5, reveal a crucial characteristic of our method. Synapse consistently outperforms the best individual model available in the pool, regardless of the ensemble size. This demonstrates that the synergistic combination of expertise within Synapse is more powerful than relying on any single, top-performing TSFM.

This robust scaling behavior is a strong indicator of the future-proof nature of Synapse. As newer and potentially more powerful TSFMs emerge, this trend suggests that Synapse will not only improve its absolute performance but might also continue to maintain an advantage over the best monolithic model.

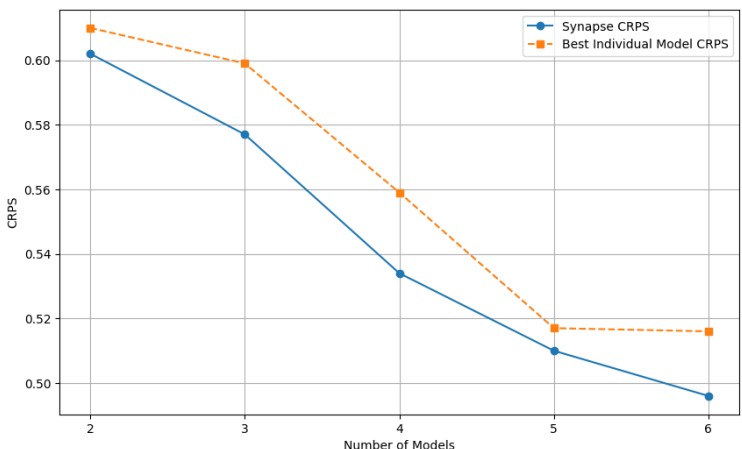

Figure 5: Performance comparison between Synapse and the best individual model in the pool. As the number of models increases, Synapse consistently achieves a lower (better) CRPS, demonstrating its effective synthesis of model expertise.

## 6 Conclusion

In this work, we identified the "representational compromise" in monolithic Time Series Foundational Models as a key barrier to robust forecasting, particularly for non-stationary data. We established an empirical performance ceiling by developing an Oracle selector, which highlighted the significant untapped potential in leveraging the complementary, time-varying expertise of existing TSFMs.

To harness this potential and close the gap to Oracle, we introduced Synapse, a novel dynamic arbitration framework. By employing a forward-simulation mechanism to dynamically adapt its weights and a predictive sampling strategy to construct a probabilistically sound forecast, Synapse effectively navigates the strengths and weaknesses of its constituent models at each timestep. Our extensive experiments demonstrate that Synapse achieves state-of-the-art performance, significantly outperforming both individual state-of-the-art models and traditional ensemble baselines. We showed that its performance advantage amplifies over longer, more challenging forecast horizons where static ensembles often falter.

Critically, we demonstrated that an arbitrated mix of weaker models, when guided by Synapse, can decisively outperform a stronger, single monolithic TSFM. This finding suggests a paradigm shift in time series forecasting: the pursuit of a single, universal model may be less fruitful than developing sophisticated methods for dynamically arbitrating an ecosystem of specialized experts.

### Limitations

While Synapse demonstrates strong performance, it's primary limitation lies in its forward simulation. The process leverages the arbitrator's own forecast as a "simulated ground truth" to update model weights. This self-referential loop risks confirmation bias, potentially favoring consensus models over correct outliers. This is particularly probable in cases, where none of the models can provide very meaningful forecasts resulting in Synapse often getting lower performance compared to that of some constituent models. Figure 7 (h) and 7 (i) shows examples where Synapse slighly underperforms constituent models. Future work may focus on breaking this loop, for instance by introducing exploration techniques to ensure that correct, non-consensus models can still gain influence.

**Acknowledgments**

We would like to thank Rui Meng, Vishy Tirumalashetty and Burak Gokturk for their valuable discussions, feedback, and guidance throughout this project.

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

## A    Experimentation Setup

We conducted all experiments using 4X NVIDIA A100 80GB GPUs. These resources were utilized primarily for deployment and execution of the TSFMs - Moirai small/medium/large, Sundial, Toto, and Moirai-2.

Initialization window $\mathcal{W}_{\text{init}}$ is established by one backtest window on the input, with the window size matching that of forecast horizon $T$, which is subsequently used for weight initialization. For the predictive sampling stage of SYNAPSE arbitration, we use a total of $N_{\text{total}} = 1500$ for the final mixture, and models contribute according to their weights for the mixture of constituent predictive distribution.

## B    Evaluation Metrics

To compare the performance of SYNAPSE to that of different baselines, we primarily use two metrics: Mean Absolute Scaled Error (MASE) and Continuous Ranked Probability Score (CRPS). These metrics are defined as below:

- **Continuous Ranked Probability Score (CRPS):** To evaluate the entire predictive distribution. CRPS measures the integrated squared difference between the forecast's cumulative distribution function (CDF) and the empirical CDF of the ground truth outcome. Exact computation of CRPS however can be computationally expensive. Therefore, weighted quantile loss (Park et al., 2022) - discrete sum over a finite set of quantile levels can be used as an approximation.

$$\text{CRPS}_{\text{approx}} = \frac{1}{K} \sum_{k=1}^{K} \text{wQL}[\alpha_k]$$

  where $K$ is the number of quantile levels (e.g., $K = 9$ for levels $\{0.1, 0.2, \ldots, 0.9\}$). The weighted quantile loss for a single level $\alpha$ is a normalized version of the pinball loss:

$$\text{wQL}[\alpha] = \frac{2 \cdot \rho_\alpha(\hat{q}_t(\alpha), y_t)}{|y_t|}$$

  Here, $y_t$ is the ground truth value, $\hat{q}_t(\alpha)$ is the predicted value for the $\alpha$-th quantile, and $\rho_\alpha$ is the standard pinball loss function, defined as:

$$\rho_\alpha(\hat{q}, y) = \begin{cases} \alpha(y - \hat{q}) & \text{if } y > \hat{q} \\ (1 - \alpha)(\hat{q} - y) & \text{if } y \le \hat{q} \end{cases}$$

  A lower CRPS indicates a more accurate and well-calibrated probabilistic forecast.

- **Mean Absolute Scaled Error (MASE):** To provide a scale-independent measure of point forecast accuracy, we use the Mean Absolute Scaled Error (MASE). MASE normalizes the forecast's Mean Absolute Error (MAE) by the in-sample MAE of a naive, one-step seasonal benchmark. This allows for meaningful comparison of accuracy across diverse time series, regardless of their original scale.

  Formally, MASE is defined as:

$$\text{MASE} = \frac{\text{MAE}_{\text{forecast}}}{\text{MAE}_{\text{naive, seasonal}}}$$

## C    Domain-Level Performance Comparison of TSFMs vs Oracle

To further quantify the potential gains from dynamic arbitration, we present a granular performance breakdown of each constituent TSFM across the distinct domains of the GIFT-Eval. As shown in Table 4, even the single best-performing model for a specific domain consistently and substantially underperforms the *Oracle Arbitrator*. This performance gap highlights the critical importance of dynamic, timestep-level arbitration over static or domain-level model selection.

Table 4: Domain-wise performance comparison (CRPS / MASE) of individual TSFMs against the Oracle Arbitrator. For each domain and metric, the **best** (lowest) performing individual TSFM is bolded, and the second best is underlined. The Oracle's performance represents the theoretical lower bound achievable by perfectly selecting the best model at each timestep. Note that the Oracle consistently outperforms even the domain-specific best models by a wide margin.

| Domain | Metric | Moirai-L | Moirai-B | Moirai-S | Sundial | Toto | Moirai-2 | Oracle |
|---|---|---|---|---|---|---|---|---|
| Econ/Fin | CRPS | 0.778 | 0.835 | 0.834 | 0.892 | 0.848 | **0.745** | **0.696** |
|  | MASE | 0.845 | 0.905 | 0.985 | 0.923 | 0.828 | **0.779** | **0.761** |
| Energy | CRPS | 0.732 | 0.725 | 0.763 | 0.645 | 0.628 | **0.622** | **0.383** |
|  | MASE | 1.026 | 0.998 | 1.069 | 0.839 | 0.876 | **0.837** | **0.519** |
| Healthcare | CRPS | 0.565 | 0.549 | 0.744 | 0.772 | 0.467 | **0.442** | **0.371** |
|  | MASE | 0.699 | 0.683 | 0.848 | 0.798 | 0.625 | **0.600** | **0.409** |
| Nature | CRPS | 0.382 | 0.388 | 0.413 | 0.361 | **0.348** | 0.368 | **0.285** |
|  | MASE | 0.750 | 0.771 | 0.807 | **0.703** | 0.736 | 0.755 | **0.562** |
| Sales | CRPS | 0.445 | 0.424 | 0.442 | 0.475 | 0.424 | **0.414** | **0.361** |
|  | MASE | 0.711 | 0.695 | 0.731 | 0.730 | 0.705 | **0.689** | **0.601** |
| Transport | CRPS | **0.451** | 0.478 | 0.551 | 0.504 | 0.477 | 0.479 | **0.360** |
|  | MASE | **0.601** | 0.637 | 0.731 | 0.634 | 0.632 | 0.620 | **0.477** |
| Web/CloudOps | CRPS | 0.748 | 0.781 | 0.777 | 0.552 | **0.500** | 0.510 | **0.301** |
|  | MASE | 1.125 | 1.257 | 1.136 | 0.695 | 0.694 | **0.665** | **0.477** |

## D   Pairwise Comparison

To evaluate the performance consistency of SYNAPSE, we conduct a head-to-head comparison against individual constituents and ensemble baselines across all 97 configurations in GIFT-Eval as shown in Figure 6. Overall SYNAPSE maintains substantially higher win percentage over other baselines. The strong overall performance numbers paired with strong win percentages effectively demonstrates the robustness and consistency of SYNAPSE in different time series prediction situations.

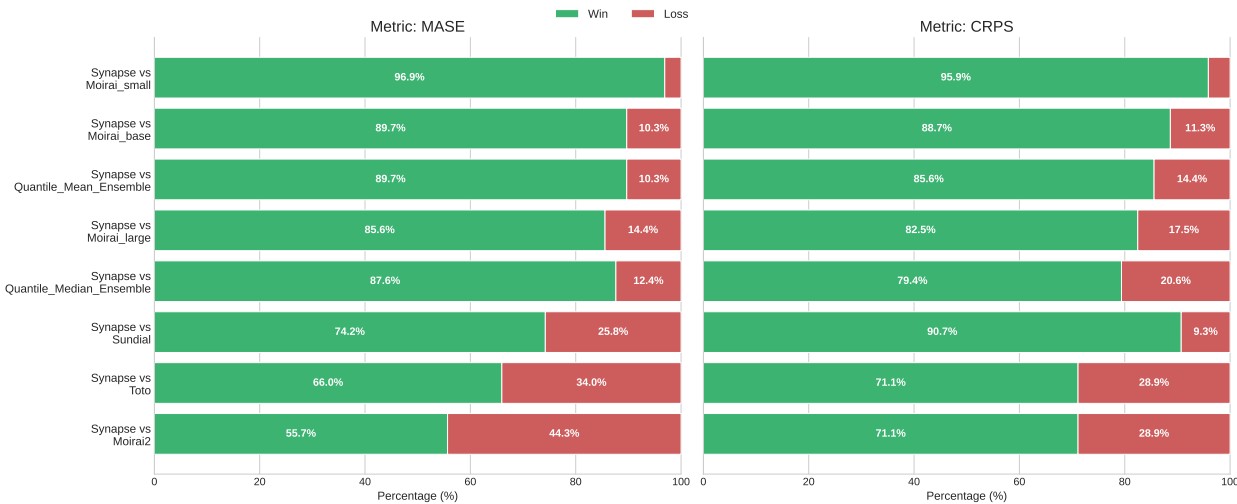

Figure 6: Pairwise Win/Loss performance comparison of SYNAPSE against constituent individual models and Quantile Median/Mean ensemble baselines. Besides strong overall performance, comparative analysis demonstrates the overall consistency of SYNAPSE.

# E  Performance Scaling with Models - Detailed Comparison

As explained in Section 5.8, we chronologically order the constituent models - Moirai-small, Moirai-base, Moirai-large, Sundial, Toto, Moirai-2 and construct model pool of increasing size for arbitration. Table 5 shows the detailed result of model performance scaling comparison with number of models vs best performing individual model.

Table 5: Performance comparison of SYNAPSE against the Best Single Model baseline. Scores are shown for different model pools. Here Moirai-S,B,L indicates a pool consisting of Moirai-small, Moirai-medium, and Moirai-large models.

| Models | CRPS | | MASE | |
|---|---|---|---|---|
| | SYNAPSE | Best Individual Model | SYNAPSE | Best Individual Model |
| Moirai-S,B | **0.602** | 0.610 | **0.882** | 0.901 |
| Moirai-S,B,L | **0.577** | 0.599 | **0.850** | 0.875 |
| Moirai-S,B,L, Sundial | **0.534** | 0.559 | 0.778 | **0.750** |
| Moirai-S,B,L, Sundial, Toto | **0.510** | 0.517 | **0.741** | 0.750 |
| Moirai-S,B,L, Sundial, Toto, Moirai-2 | **0.496** | 0.516 | **0.719** | 0.728 |

# F  Inference Time Complexity Analysis

SYNAPSE works as a post-hoc arbitration mechanism, that executes all $N$ constituent Time Series Foundation Models (TSFMs) exactly once in parallel to generate forecasts for the entire horizon $T$. Subsequently, the "forward simulation" recalculates scalar weights $\{w_{i,t}\}$ by sampling from pre-computed distributions; no arbitrated output is fed back into a TSFM for re-inference. Consequently, the total time complexity is the sum of the parallel inference latency of the constituent models and the sequential arbitration logic. The time complexity is formally decomposed as:

$$\mathcal{C}_{\text{total}} = \mathcal{O}\left(\max_{i=1}^{N}(\text{Inference}(M_i))\right) + \mathcal{O}(\eta \cdot T) \tag{3}$$

where $\eta$ represents the constant overhead of the arbitration logic per step. Transformer based TSFM inference typically involves quadratic attention complexity $\mathcal{O}(\tau^2)$, and the arbitration overhead scales linearly with the forecast horizon $\mathcal{O}(T)$.

Empirically, we measure the total wall-clock time for SYNAPSE against the slowest constituent model (the parallel execution floor) across different datasets and horizons. As detailed in Table 6, the results confirm that the base inference cost serves as a floor (with complex models like Toto on standard datasets), while the forward simulation introduces an additive, linear overhead.

Table 6: Wall-clock inference time comparison (in seconds). The "Max Base Latency" represents the time taken by the slowest individual TSFM in the pool (parallel execution floor). The "SYNAPSE Overhead" isolates the additional time required for the dynamic arbitration logic.

| Dataset | Horizon ($T$) | Max Base Latency (s) | Synapse Total (s) | Synapse Overhead (s) |
|---|---|---|---|---|
| | 48 | | ∼25 | ∼+10 |
| ETT1/H | 480 | 10-15 | ∼38 | ∼+20 |
| | 720 | | ∼52 | ∼+37 |
| KDD_Cup_2018/D | 30 | 11 | 23 | +12 |
| Car Parts | 12 | 22 | 44 | +22 |
| M4 Yearly | 6 | 120 | 179 | +59 |

In short-horizon scenarios (e.g., ETT1 Short), the arbitration overhead is relatively small in comparison to base latency. As the forecast horizon $T$ increases (see ETT1 Medium/Long), SYNAPSE overhead grows

linearly in comparison to the base latency. Note that M4 Yearly is substantially larger than other datasets here. Here, this additive latency is the necessary computational investment to enable dynamic arbitration. SYNAPSE utilizes this linear time cost to generate the simulated ground truth that drives its superior predictive performance while remains practical even for extended horizons.

## G  Synapse: Forecast Stability

SYNAPSE arbitrates on constituent TSFM forecasts solely by sampling from their predictive distributions post-hoc and dynamically adjusting weights based on scoring rules. Consequently, the framework exhibits high forecast stability. To quantify this, we executed SYNAPSE five times on a subset of the GIFT-Eval benchmark. As reported in Table 7, the standard deviation of the performance metrics across these runs is quite low, showing that the arbitration mechanism results in overall stable performance.

Table 7: Forecast stability analysis of SYNAPSE over 5 independent runs. The extremely low standard deviations for both MASE and CRPS demonstrate that the arbitration mechanism is robust to sampling noise.

| Dataset (Domain) | Horizon | MASE (Mean ± Std. Dev) | CRPS (Mean ± Std. Dev) |
|---|---|---|---|
| Car Parts (Sales) | Short | $0.8361 \pm 0.0001$ | $0.9543 \pm 0.0002$ |
| ETT1/H (Energy) | Long | $1.3431 \pm 0.0001$ | $0.2608 \pm 0.00005$ |
| ETT1/H (Energy) | Medium | $1.2440 \pm 0.0002$ | $0.2497 \pm 0.00003$ |
| ETT1/H (Energy) | Short | $0.8193 \pm 0.0004$ | $0.1778 \pm 0.0001$ |
| KDD Cup 2018/D (Nature) | Short | $1.1753 \pm 0.0002$ | $0.3732 \pm 0.0001$ |

## H  Comparison with Mixture of Experts (MoE) Model

While SYNAPSE is conceptually related to "dynamic specialization" with Mixture of Experts architectures like Moirai-MoE (Liu et al., 2024), the two approaches address fundamentally different constraints. MoE models are typically trained end-to-end while SYNAPSE is a *post-hoc arbitration* framework designed to arbitrate over pre-existing, frozen foundational models. This distinction allows SYNAPSE to be model-agnostic; it can arbitrate between any set of forecasters, including MoE models themselves, thereby leveraging the strongest available tools without the prohibitive computational cost of retraining specialized experts from scratch.

Moirai-MoE results are not reported to GIFT-Eval leaderboard, making it more difficult to compare the performance against SYNAPSE. Furthermore, Moirai-MoE is trained on LOTSA (Woo et al., 2024) creating a potential data leakage issue for several datasets in GIFT-Eval, making a strictly fair zero-shot comparison difficult. Nevertheless, their reported unleaked zero-shot results on the BizITObs (Web/Cloud) dataset offers a comparative perspective. Here, SYNAPSE demonstrates substantially stronger performance compared to Moirai-MoE. Specifically, SYNAPSE achieves a **MASE of 0.264** and **CRPS of 0.072**, decisively outperforming both Moirai-MoE Small (MASE 0.298, CRPS 0.081) and Moirai-MoE Base (MASE 0.290, CRPS 0.079). These results suggest that post-hoc arbitration can effectively rival or surpass the performance of end-to-end specialized architectures in zero-shot scenarios, while maintaining the flexibility to incorporate such models as constituents in the future.

## I   Performance Scaling of Synapse with Architecturally Diverse TSFMs

Table 8: Performance scaling with most recent SOTA TSFMs. SYNAPSE arbitrates over a diverse pool including FlowState (SSM), Chronos 2 (Encoder-based), and TimesFM 2.5 (Decoder-based), consistently outperforming the single best model.

| Model / Method | MASE | CRPS |
|---|---|---|
| Toto | 0.750 | 0.517 |
| Moirai 2 | 0.728 | 0.516 |
| FlowState | 0.726 | 0.502 |
| TimesFM-2.5 | 0.705 | 0.490 |
| Chronos-2 | 0.698 | 0.485 |
| SYNAPSE (w/o Chronos 2) | 0.699 | 0.479 |
| SYNAPSE (All 5 Models) | **0.693** | **0.474** |

To verify that SYNAPSE scales effectively with model capability, we extend our evaluation to include a pool of most recently proposed SOTA TSFMs. Each model in this enhanced pool are architecturally more distinct: **Chronos-2** (Ansari et al., 2025) is an encoder-only transformer model that frames forecasting as a language translation task via tokenization; **TimesFM-2.5** employs patched decoder-only architecture to efficiently process massive contexts; **FlowState** (Graf et al., 2025) utilizes State Space Models to generate predictions; **Toto** (Cohen et al., 2024) integrates a Student-T mixture head to robustly model heavy-tailed distributions; and **Moirai-2** (Liu et al., 2025a) implements a multi-token autoregressive decoder for inference speed. As detailed in Table 8, SYNAPSE successfully arbitrates among these different architectures to produce a result superior to any single constituent model. Specifically, the full 5-model configuration achieves a MASE of **0.693** and CRPS of **0.474**, surpassing the strongest individual model in the pool (Chronos-2) by leveraging the complementary strengths. Notably, SYNAPSE **without** Chronos-2 achieves comparable MASE and better CRPS than Chronos-2 alone. This confirms that irrespective of how base models improve, SYNAPSE adds value beyond those updates, ensuring the framework continues to set new performance standards.

## J   Qualitative Plots

To offer a qualitative perspective of SYNAPSE and its performance trends in challenging forecast scenarios, we plot SYNAPSE forecasts to that of constituent baselines. Figure 7 demonstrates that SYNAPSE follows the ground truth more closely than the constituents. In simpler scenarios where most models perform competitive performance, SYNAPSE improvements are more modest, whereas challenging examples where models tend to disagree more, performance improvements are more substantial (e.g. M4_Weekly/short). We also showcase showcase some less common cases in Figure 7 (h) and 7 (i) from Nature domain, where SYNAPSE MASE was slightly higher than some of the constituent models. Upon closer investigation, we can see that, when none of the models in the pool can give meaningful forecast, SYNAPSE cannot offer any improvement and may often lag behind slightly.

Figure 7: Qualitative forecasting results comparing SYNAPSE with baselines on the best-performing series from 7 different domains.

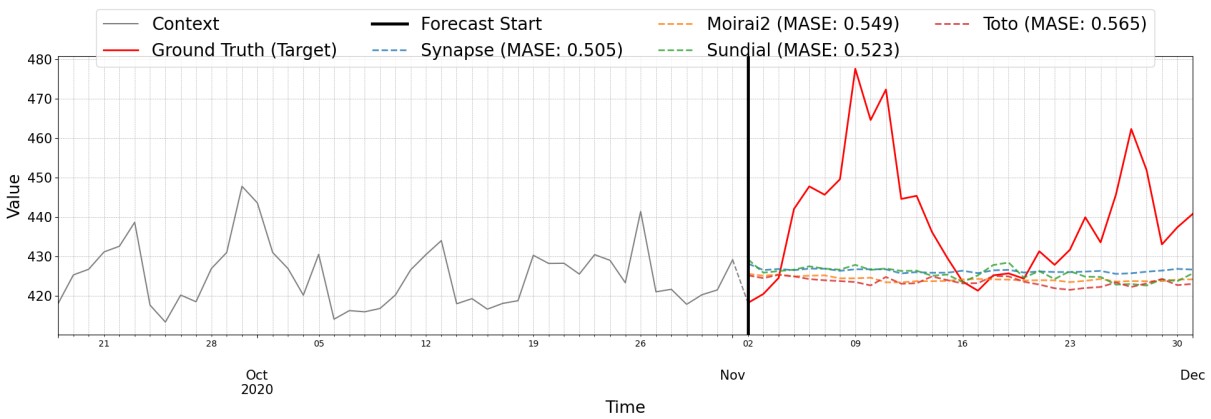

(a) Nature: Jena Weather/D/short ID 40

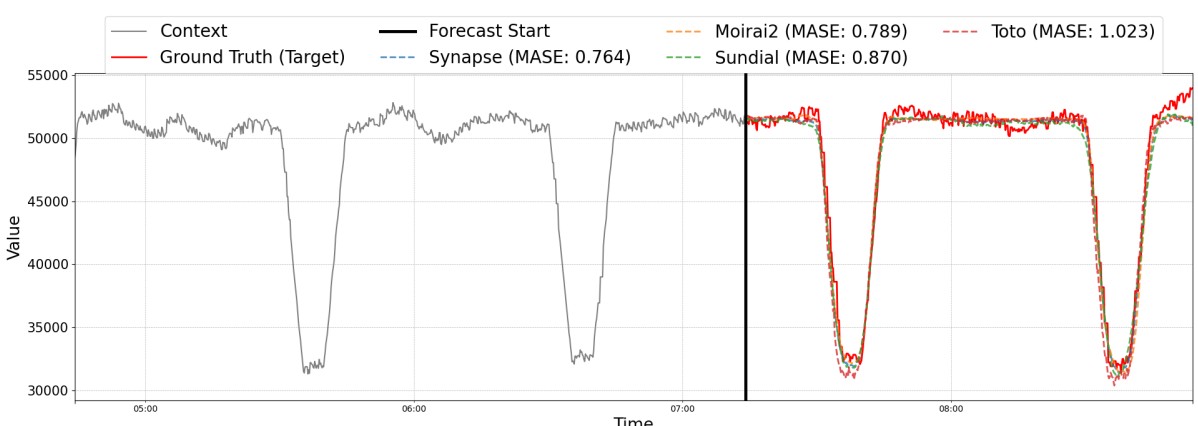

(b) Web/CloudOps: BizITObs - Application/10S/medium ID 2

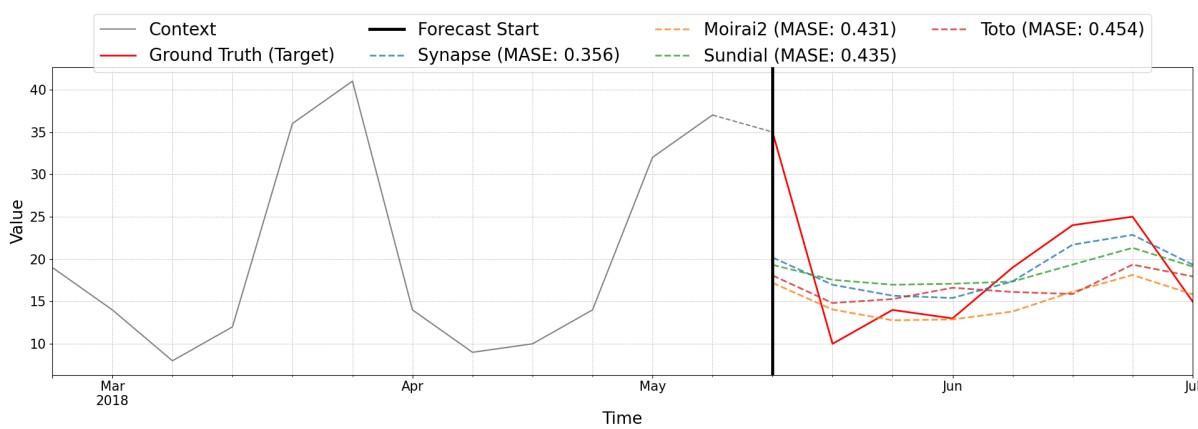

(c) Sales: Hierarchical Sales/W/short ID 88

(Continued) Qualitative forecasting results.

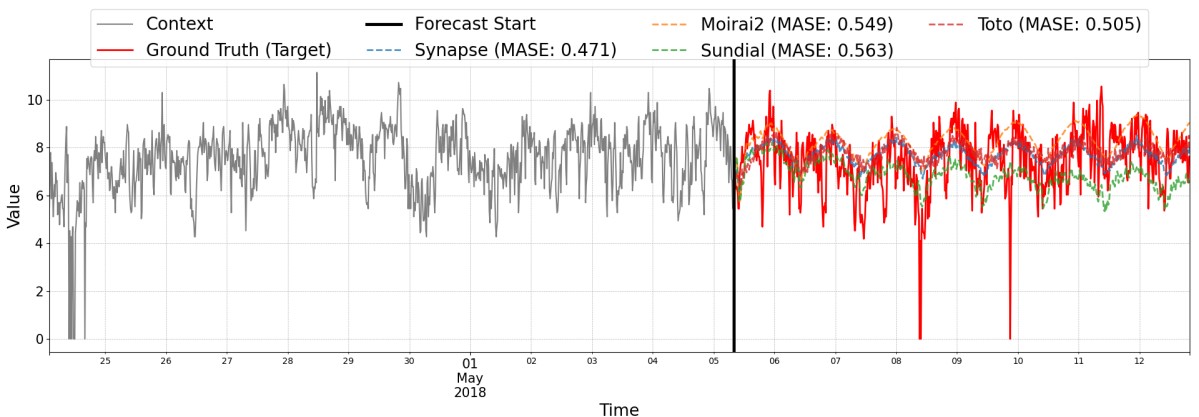

(d) Energy: ETT2/15T/long ID 13

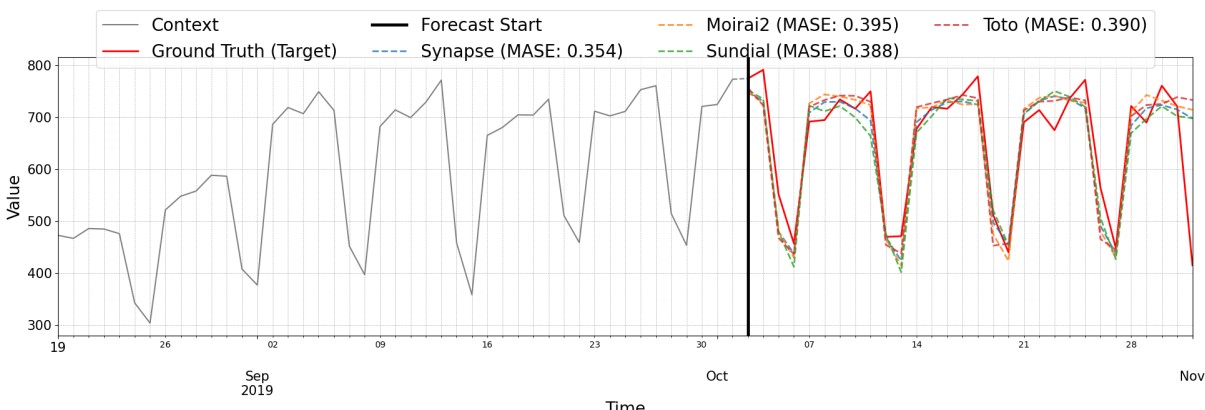

(e) Transport: M_DENSE/D/short ID 51

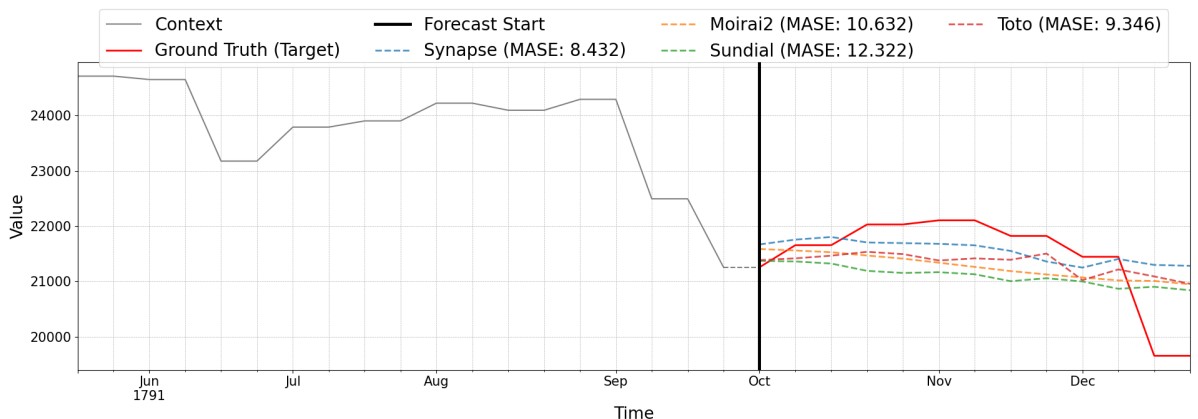

(f) Econ/Fin: M4 Weekly/short ID 248

(Continued) Qualitative forecasting results.

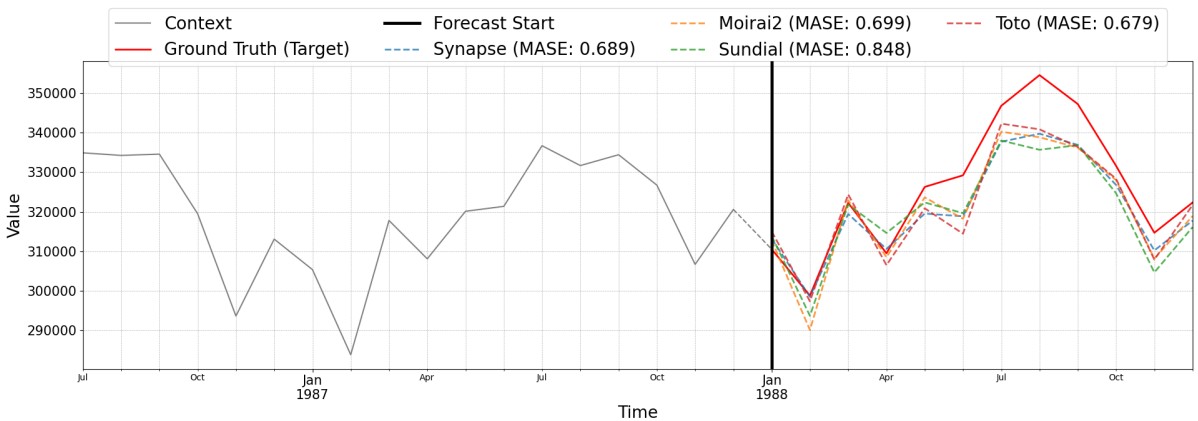

(g) Healthcare: US Births/M/short ID 1

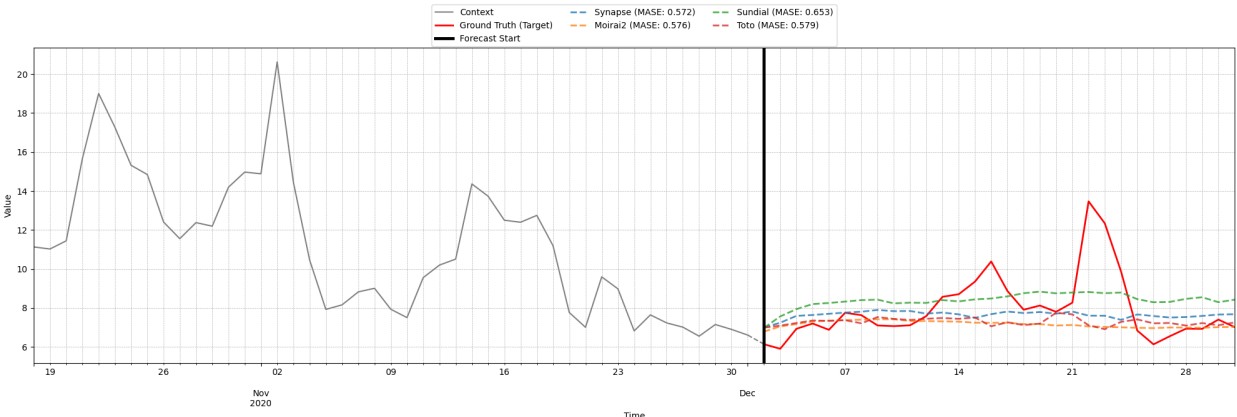

(h) Nature: Jena Weather/D/short ID 11

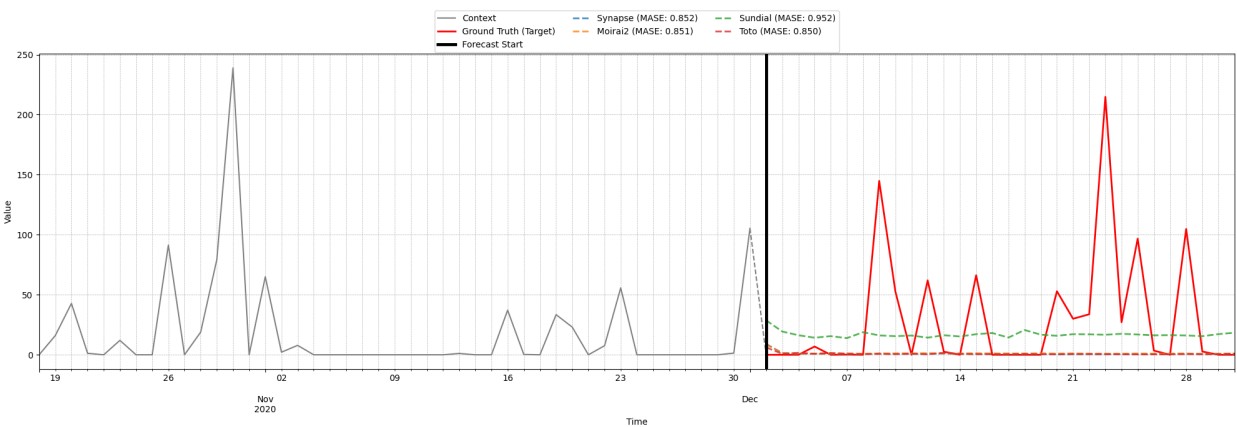

(i) Nature: Jena Weather/D/short ID 31

