# OpenReview forum: "Synapse: Adaptive Arbitration of Complementary Expertise in Time Series Foundational Models"
_TMLR — Accepted by TMLR_

### Review · Reviewer_K563 · 2025-12-14

**Summary Of Contributions:**

This paper first identifies the representational compromise inherent in monolithically trained models as a key limitation in non-stationary time-series forecasting. To address this issue, the authors reformulate the forecasting task and introduce a dynamic arbitration problem among time-series forecasting models (TSFMs). Furthermore, they propose SYNAPASE, a novel method capable of dynamically selecting a small subset of models and adaptively arbitrating their contributions at each time step. Extensive experimental results demonstrate that SYNAPASE consistently achieves state-of-the-art performance with substantial improvements over existing methods.

**Audience:**

Yes

**Audience Explanation:**

It is relevant to time series data analysis community.

**Broader Impact Concerns:**

No broader impact concerns.

**Claims And Evidence:**

Yes

**Claims Explanation:**

Yes the claim is supported by experimental results and corresponding analysis.

**Requested Changes:**

1. In table 3, the authors are suggested to add the error bars for the major results. Statistic significant results are very important to justify the conclusion.

2. The authors have only compared their methods with simple ensemble methods, e.g., mean ensemble and median ensemble, how about the performances of mixture of experts methods? (e.g., some MOESs also adopte expert choice routine [1])

[1] Zhou Y, Lei T, Liu H, et al. Mixture-of-experts with expert choice routing[J]. Advances in Neural Information Processing Systems, 2022, 35: 7103-7114.

3. Lack of discussion regarding the efficiency of the proposed approach. More details regarding the training and inference overheads should be described in Table 1, e.g., number of parameters, GFLOPS and inference time.


4. The authors are encouraged to add more failure case analysis in the paper to enbale more insights discussion regarding the limitation of the proposed approach.

5. TSNE visualizations of the embedding spaces of different models are very interesting to be demonstrated.

---

> ### Author Response · Authors · 2026-01-09
> **Author Response to Reviewer K563 (1/2)**
>
> We sincerely thank you for your constructive suggestions. We have revised the manuscript to include the requested statistical analysis, additional comparisons with SOTA/MoE models, and a detailed complexity analysis. Below we address your specific comments:
>
> > **“In table 3, the authors are suggested to add the error bars for the major results. Statistic significant results are very important to justify the conclusion.”**
>
> Thank you so much for putting emphasis on statistical significance to justify our conclusions. While re-running the entire GIFT-Eval on the complete 23-dataset benchmark multiple times is computationally prohibitive, we conducted a rigorous stability analysis ($N=5$ runs) on a subset of datasets spanning the Energy (ETT1), Sales (Car Parts), and Nature (KDD Cup 2018) domains.
>
> As demonstrated in the table below (and **Appendix G** of the revised manuscript), the standard deviations of MASE and CRPS are very low, effectively negligible and orders of magnitude smaller than the performance gaps between Synapse and the baselines. This confirms that the arbitration mechanism is highly robust, ensuring that the single-run results reported in the main paper accurately reflect the method's true performance without significant variance.
>
> **Stability Analysis Results (Table 7):**
>
> | Dataset | Domain | Horizon | MASE (Mean ± Std. Dev) | CRPS (Mean ± Std. Dev) |
> | :--- | :--- | :--- | :--- | :--- |
> | Car Parts | Sales | Short | $0.8361 \pm 0.0001$ | $0.9543 \pm 0.0002$ |
> | ETT1 | Energy | Long | $1.3431 \pm 0.0001$ | $0.2608 \pm 0.00005$ |
> | ETT1 | Energy | Medium | $1.2440 \pm 0.0002$ | $0.2497 \pm 0.00003$ |
> | ETT1 | Energy | Short | $0.8193 \pm 0.0004$ | $0.1778 \pm 0.0001$ |
> | KDD Cup 2018 | Nature | Short | $1.1753 \pm 0.0002$ | $0.3732 \pm 0.0001$ |
>
> > **“The authors have only compared their methods with simple ensemble methods... how about the performances of mixture of experts methods?”**
>
> Thank you for the suggestion. Unlike MoEs like Moirai-MoE [1] for time series forecasting, which require end-to-end training, Synapse performs post-hoc arbitration of frozen models, allowing it to flexibly leverage any architecture (including MoEs themselves) without training models from scratch.
>
> While direct comparison on GIFT-Eval is restricted by missing baselines and potential data leakage in MoE training corpora (LOTSA), we have added **Appendix H** to provide a fair zero-shot comparison on the verified unleaked **BizITObs** dataset. In this setting, Synapse (MASE 0.264, CRPS 0.072) outperforms both Moirai-MoE Small (MASE 0.298, CRPS 0.081) and Base (MASE 0.290, 0.079), demonstrating that dynamic arbitration can rival end-to-end learned routing strategies.
>
> > **“Lack of discussion regarding the efficiency of the proposed approach. More details regarding the training and inference overheads should be described...”**
>
> Thank you for your suggestions on computational efficiency and inference overhead. We have expanded our analysis in the revised manuscript to address this. Regarding “training” overhead, we clarify that Synapse is a training-free, post-hoc framework; unlike end-to-end single model approaches, it requires zero gradient updates and introduces no additional trainable parameters.
>
> Regarding inference, we have added a dedicated **"Inference Time Complexity Analysis"** in **Appendix F**, detailing timings and formal complexity. We demonstrate that the total time complexity is the sum of the inference time of the constituent models plus a linear arbitration term: $\mathcal{C}_{total}$ =
>
> $\mathcal{O}(\max_{i=1}^{N}(\mathrm{Inference}(M_i))) + \mathcal{O}(\eta T)$
>
> Our empirical results in **Appendix F** show that for short horizons, the arbitration overhead ($\eta$) is negligible compared to the base model latency. While this overhead grows linearly with the forecast horizon $T$, it represents a deliberate trade-off where Synapse utilizes this simulation cost to drive superior accuracy, significantly widening the performance gap against baselines in long-horizon tasks.
>
> > **“The authors are encouraged to add more failure case analysis in the paper to enbale more insights discussion regarding the limitation of the proposed approach.”**
>
> We appreciate your suggestions about adding more failure case analysis. We have added some failure cases in the **Qualitative Plots** of **Appendix J**, where Synapse slightly lags behind some of the constituent models. From the added plots, we can notice that in challenging cases, when none of the constituent models can give meaningful predictions, Synapse cannot improve performance, and often underperforms compared to some of the constituents. We have also updated the **Limitations** section to discuss this "consensus bias" risk.

---

> ### Author Response · Authors · 2026-01-09
> **Author Response to Reviewer K563 (2/2)**
>
> > **“TSNE visualizations of the embedding spaces of different models are very interesting to be demonstrated.”**
>
> Thank you for the interesting suggestion to visualize the embedding spaces to demonstrate model distinctness. However, a direct t-SNE comparison may not be straightforward in this context because the constituent models utilize fundamentally incompatible architectures with disjoint latent spaces. More specifically, models from the Moirai family are encoder-based, Moirai2 and Toto employ decoder-based architectures, and Sundial uses flow matching lacking discrete tokenization.
>
> To quantitatively address the underlying goal of demonstrating diversity, we instead provide rigorous **Oracle Diagnostics** in **Figure 1 (Right)**, where high Selection Entropy ($>2.4$ bits) and low Modal Shares ($<30$%) confirm that the models exhibit distinct behaviors and disjoint areas of expertise across the input space, effectively validating the functional diversity that an embedding visualization would aim to show.
>
> **Updated Oracle Diagnostics (Summary Table):**
>
> | Domain | Horizon | Modal (Oracle) | Selection Entropy (Bits) | Modal Share | Switch Freq. % |
> | :--- | :--- | :--- | :---: | :---: | :---: |
> | Econ/Fin | Short | Sundial | 2.541 | 0.251 | 14.47% |
> | Energy | Long | Moirai2 | 2.432 | 0.281 | 45.31% |
> | Energy | Medium | Sundial | 2.421 | 0.288 | 44.86% |
> | Energy | Short | Sundial | 2.467 | 0.278 | 43.53% |
> | Healthcare | Short | Sundial | 2.531 | 0.237 | 41.28% |
> | Nature | Long | Sundial | 2.505 | 0.283 | 29.36% |
> | Nature | Medium | Sundial | 2.497 | 0.285 | 27.79% |
> | Nature | Short | Moirai | 2.544 | 0.213 | 40.82% |
> | Sales | Short | Sundial | 2.550 | 0.242 | 47.27% |
> | Transport | Long | Moirai | 2.494 | 0.278 | 51.75% |
> | Transport | Medium | Moirai | 2.529 | 0.254 | 51.08% |
> | Transport | Short | Sundial | 2.550 | 0.236 | 58.45% |
> | Web/CloudOps | Long | Moirai2 | 2.562 | 0.225 | 49.31% |
> | Web/CloudOps | Medium | Moirai2 | 2.556 | 0.223 | 50.09% |
> | Web/CloudOps | Short | Toto | 2.509 | 0.240 | 49.67% |
>
> [1] Liu X, Liu J, Woo G, Aksu T, Liang Y, Zimmermann R, Liu C, Savarese S, Xiong C, Sahoo D. Moirai-moe: Empowering time series foundation models with sparse mixture of experts. arXiv preprint arXiv:2410.10469. 2024 Oct 14.

---

### Review · Reviewer_jvut · 2025-12-18

**Summary Of Contributions:**

- The authors introduce Synapse, a tool for dynamically utilizing a pool of TSFMs, assigning and adjusting prediction weights based on their relative, context-dependent performance, and creating a robust prediction distribution by adaptively sampling from the output quantiles of the individual models.
- I like figure 1, and it serves as motivation (how often to change the model for the best possible prediction). The paper could be strengthened by adding how great the performance loss would be without the many changes.
- The idea presented sounds logical, interesting, and shows good results.
- I like Figure 2. It reinforces the authors' motivation, but I also find the comparison somewhat unfair at this point, as one model is being compared with the best possible prediction out of different models (oracle). I think that an ensemble would be more appropriate or should be added.
- I find the paper generally easy to understand. The dataset and models seem appropriate. Sections 4.5 and 4.6 substantiate the paper.

**Additional Comments:**

- Is there no research on dynamic arbitration? Is the idea of going beyond uniform distribution, as in the ensemble, unique to this paper?

**Audience:**

Yes

**Audience Explanation:**

Yes, I think it's an interesting and timely topic.

**Claims And Evidence:**

Yes

**Claims Explanation:**

In general, the claims were substantiated by reasoning and experiments, but I have two points that I find insufficient:

- "This persistent and high-frequency switching underscores that no single model is optimal for an entire forecast window and provides a strong empirical justification for our dynamic arbitration approach, as a static model selection would be consistently suboptimal." I understand the motivation here, and in particular that one would like to have a model that predicts as accurately as possible across all data. However, I am missing a comparison where each model was evaluated individually on each domain and horizon, without merging results, in order to justify the statement. It would be possible to determine which model performs best on small subsets of the data and then use that model.

- "This demonstrates that the synergistic combination of expertise within Synapse is more powerful than relying on any single, top-performing TSFM." & "Critically, we demonstrated that an arbitrated mix of weaker models, when guided by Synapse, can decisively outperform a stronger, single monolithic TSFM" I don't think this statement was shown here, because Figure 6 does not compare against a stronger model, but against the strongest in the pool currently being considered (i.e., a model that is also part of Synapse). According to Figure 6, Toto and Moirai-2 alone would be stronger than Synapse using the four weaker models.

**Requested Changes:**

Critical changes:
- I lack a comparison of how computationally intensive Synapse is compared to baselines, especially in terms of inference time.
- I understand the difference to MoE methods, as these are learned end-to-end, but the motivation is similar and, especially during inference, they are more efficient. I would also add an approach as a baseline.
- In some cases, it makes sense to put related work at the end, but I wouldn't do that here, as it would help the reader understand at the beginning and directly clarifies why these models were chosen.
- The qualitative plots in Appendix D should be described.
- It was mentioned in the appendix that N was set to 1500. I am missing an ablation at this point, as this value seems high to me, and a comparison of the inference times would also be interesting.


Minor changes:
- I would add Oracle to Table 1 to see how big the performance gap is.
- Figures 4 and 5 merely present the results from Table 1 in a different way. I would make better use of the space in the main paper and move these plots to the appendix.
- References to the appendix would be useful, not just to B.
- The labelling of the plots should be enlarged.

---

> ### Author Response · Authors · 2026-01-09
> **Author Response to Reviewer jvut (1/2)**
>
> We sincerely thank you for your thoughtful and constructive feedback. The manuscript has been revised to include additional experiments and improved presentation based on these insights. Detailed responses to each point are provided below.
>
> > **“This persistent and high-frequency switching underscores that no single model is optimal for an entire forecast window and provides a strong empirical justification for our dynamic arbitration approach, as a static model selection would be consistently suboptimal." I understand the motivation here, and in particular that one would like to have a model that predicts as accurately as possible across all data. However, I am missing a comparison where each model was evaluated individually on each domain and horizon, without merging results, in order to justify the statement. It would be possible to determine which model performs best on small subsets of the data and then use that model.”**
>
> Thank you very much for the suggestion. We have included a comprehensive, unmerged domain-level performance breakdown for each TSFM in **Appendix C** of the revised manuscript. For your convenience, we have also reproduced this table below.
>
> **Domain-Level Performance (Table 4):**
>
> | Domain | Metric | Moirai-L | Moirai-B | Moirai-S | Sundial | Toto | Moirai-2 | Oracle |
> | :--- | :--- | :--- | :--- | :--- | :--- | :--- | :--- | :--- |
> | **Econ/Fin** | CRPS | 0.778 | 0.835 | 0.834 | 0.892 | 0.848 | **0.745** | 0.696 |
> | | MASE | 0.845 | 0.905 | 0.985 | 0.923 | 0.828 | **0.779** | 0.761 |
> | **Energy** | CRPS | 0.732 | 0.725 | 0.763 | 0.645 | 0.628 | **0.622** | 0.383 |
> | | MASE | 1.026 | 0.998 | 1.069 | 0.839 | 0.876 | **0.837** | 0.519 |
> | **Healthcare** | CRPS | 0.565 | 0.549 | 0.744 | 0.772 | 0.467 | **0.442** | 0.371 |
> | | MASE | 0.699 | 0.683 | 0.848 | 0.798 | 0.625 | **0.600** | 0.409 |
> | **Nature** | CRPS | 0.382 | 0.388 | 0.413 | 0.361 | **0.348** | 0.368 | 0.285 |
> | | MASE | 0.750 | 0.771 | 0.807 | **0.703** | 0.736 | 0.755 | 0.562 |
> | **Sales** | CRPS | 0.445 | 0.424 | 0.442 | 0.475 | 0.424 | **0.414** | 0.361 |
> | | MASE | 0.711 | 0.695 | 0.731 | 0.730 | 0.705 | **0.689** | 0.601 |
> | **Transport** | CRPS | **0.451** | 0.478 | 0.551 | 0.504 | 0.477 | 0.479 | 0.360 |
> | | MASE | **0.601** | 0.637 | 0.731 | 0.634 | 0.632 | 0.620 | 0.477 |
> | **Web/Cloud** | CRPS | 0.748 | 0.781 | 0.777 | 0.552 | **0.500** | 0.510 | 0.301 |
> | | MASE | 1.125 | 1.257 | 1.136 | 0.695 | 0.694 | **0.665** | 0.477 |
>
> As shown, even the best-performing model within a specific domain substantially underperforms the Oracle arbitrator, which underscores the necessity of timestep-level arbitration. Furthermore, we have expanded Section 3.3 to include a deeper analysis of the Oracle arbitrator-specifically examining modal models, modal share, selection entropy, and switching frequency.
>
> > **“'This demonstrates that the synergistic combination... is more powerful than relying on any single, top-performing TSFM.'... I don't think this statement was shown here, because Figure 6 does not compare against a stronger model...”**
>
> Thank you very much for the pointer. We agree that our original phrasing here may not give a clear picture given that specific context.
>
> However, we would like to highlight our new experiment added in **Appendix I (Table 8)** using a distinct set of newer and diverse SOTA models (**Chronos-2, TimesFM-2.5, FlowState**). In this setting, we observed that Synapse without the strongest model (Chronos-2) achieved a CRPS of 0.479, effectively outperforming Chronos-2 alone (CRPS 0.485) and achieving comparable MASE (0.699 vs. 0.698).
>
> Nevertheless, to avoid confusion and overgeneralization, we have fixed the contributions part of our manuscript accordingly to focus on the synthesis of complementary expertise and improvement upon the capabilities of its individual constituent model.
>
> > **“I lack a comparison of how computationally intensive Synapse is compared to baselines, especially in terms of inference time.”**
>
> We have added a dedicated **"Computational Complexity Analysis"** section in **Appendix F** of the revised manuscript (Table 6) of some datasets in GIFT-Eval, which provides detailed inference timings and a formal complexity analysis. The total time complexity $\mathcal{C}_{total}$=
>
> $$
> \mathcal{O}\left(\max_{i=1}^{N}(\mathrm{Inference}(M_i))\right) + \mathcal{O}(\eta T)
> $$
>
> $\eta$ represents constant overhead of arbitration per step. Our inference time analysis indicates that arbitration overhead remains marginal relative to base latency in short-horizon scenarios; it becomes more pronounced at longer horizons. This reflects a clear accuracy-latency trade-off: Synapse leverages the linear computational cost of forward simulation to drive substantial accuracy gains, effectively widening the performance gap against baselines in long-horizon settings compared to baselines.

---

> ### Author Response · Authors · 2026-01-09
> **Author Response to Reviewer jvut (2/2)**
>
> > **“I understand the difference to MoE methods... but the motivation is similar and, especially during inference, they are more efficient. I would also add an approach as a baseline.”**
>
> Thank you very much for the suggestion. We agree that MoE architectures share the motivation of sub-task specialization. As noted in our Introduction, the key distinction is that Synapse focuses on the post-hoc arbitration of "frozen" black-box models, avoiding the training of experts from scratch. Furthermore, due to the post-hoc nature of Synapse, any model including MoE models can be used for arbitration to get improved performance over all constituents.
>
> For baseline comparison, we faced two practical challenges: (1) GIFT-Eval does not report results on Mixture of Experts models like Moirai-MoE  [1], and (2) potential data leakage in the training corpora (LOTSA) of Moirai-MoE makes fair zero-shot comparison on GIFT-Eval infeasible. Nevertheless, their reported unleaked zero-shot results on the BizITObs (Web/Cloud) dataset offer a comparative perspective.
>
> We have added **Appendix H**, where we provide this discussion as well as a targeted comparison on the BizITObs (Web/Cloud) dataset in comparable setting, which serves as a clean, unleaked evaluation ground for all models. We find that Synapse (MASE **0.264** / CRPS **0.072**) outperforms both Moirai-MoE Small (0.298 / 0.081) and Base (0.290 / 0.079) in this setting. This suggests that Synapse’s ability to dynamically ensemble diverse architectures can outperform the efficacy of end-to-end learned routing.
>
> > **“In some cases, it makes sense to put related work at the end, but I wouldn't do that here...”**
>
> Thank you very much for suggesting the reorganization of related works. As per your suggestions, we have moved Related Works after the Introduction. We hope that improves the overall readability of the manuscript.
>
> > **“The qualitative plots in Appendix D should be described.”**
>
> We have included a brief prologue to the **Qualitative Plots** section of the Appendix (**Appendix J**) to offer a brief introduction and overview of the plots. We believe this should also improve the presentation of the findings.
>
> > **“It was mentioned in the appendix that N was set to 1500. I am missing an ablation at this point, as this value seems high to me...”**
>
> We understand that the role of the parameter $N_{total}$ might be confusing. Please note that in Synapse, $N_{total}$ only determines the resolution of the aggregated distribution constructed via Cubic Spline interpolation. For example, with an arbitrated pool of 6 models having equal weights, setting $N_{total}=1500$ implies random sampling 250 points from each model’s distribution quantiles to form the final mixture. This step is a lightweight post-processing operation.
>
> To explicitly address the concern regarding efficiency, we measured the inference latency with varying $N_{total}$. Reducing $N_{total}$ from 1500 to 600 (i.e., 100 samples per model when equal weight) reduced the total inference time from 38 seconds to 37 seconds, a negligible improvement of only 1 second. This confirms that $N_{total}$ has strictly no meaningful impact on runtime complexity; it is tuned solely to ensure high-fidelity representation of the predictive distribution within the spline interpolation step.
>
> **Other Comments**
>
> As per your suggestions, we have added Oracle performances in **Table 1**, and we believe it is a very good addition to show the performance possible from an Oracle arbitrator. We have also moved the Pairwise comparison plot from the main manuscript to **Appendix D**. Finally, we have enlarged the labels of some of the figures.
>
> > **“Is there no research on dynamic arbitration? Is the idea of going beyond uniform distribution, as in the ensemble, unique to this paper?”**
>
> While static combination methods like median ensembles have been explored [2], to the best of our knowledge, Synapse is the first to investigate this property of Time Series Foundational Models from the perspective of arbitration, specifically addressing the transient nature of their expertise across forecast horizons.
>
> [1] Liu X, Liu J, Woo G, Aksu T, Liang Y, Zimmermann R, Liu C, Savarese S, Xiong C, Sahoo D. Moirai-moe: Empowering time series foundation models with sparse mixture of experts. arXiv preprint arXiv:2410.10469. 2024 Oct 14.
>
> [2] Garza A, Rosillo R. TimeCopilot. arXiv preprint arXiv:2509.00616. 2025 Aug 30.

---

### Review · Reviewer_3NJQ · 2025-12-26

**Summary Of Contributions:**

The paper evaluates time-series foundation models (TSFMs) in a post-hoc combination scheme. The authors motivate this by stating—and showing (at least under their chosen set of baselines)—that no single TSFM dominates across datasets, domains, and forecast horizons. They also propose an oracle selector that automatically chooses the best model, serving as an upper bound.

Their main method, Synapse, is an adaptive ensemble/mixture procedure. It constructs an aggregated predictive distribution via weighted predictive sampling from each TSFM’s quantile outputs, using an estimated inverse CDF derived from the quantiles.

The evaluation is conducted on the popular foundation-model benchmark GIFT-Eval, and the paper compares against a small number of foundation models, mostly based on similar architectures. Stronger current baselines, such as FlowState or Chronos 2, are omitted.

----

**Strengths**:

The paper addresses an important question: how can we leverage multiple foundation models that are individually insufficient, without requiring training a new model?

Predictive sampling yields a distribution that is probabilistically valid by construction (quantiles computed from an empirical sample are monotone), avoiding ad hoc quantile-crossing fixes.

Within its limited scope, the paper provides a solid analysis and ablation study.


**Weaknesses**:

The claim that no single model excels is plausible, but a more thorough analysis of the best-performing methods on GIFT-Eval would strengthen it.

Claims of state-of-the-art performance are not convincingly supported given the limited baseline set and the absence of stronger ensemble/combination baselines.

Several important implementation details (e.g., window size, initialization, and sensitivity analyses) are underspecified, weakening reproducibility, and there is no released reproduction code.

**Additional Comments:**

Naming: "arbitration framework" is not standard ML forecasting terminology for what is essentially a dynamic ensemble/mixture; consider renaming it or justifying the term with precedent.

Missing space:
series foundation models(Das et al., 2024; Graf et al., 2025; Woo et al., 2024; Cohen et al., 2025; Shi et al., 2024).

Toto (Cohen et al., 2024), Sundial(Liu et al., 2025b), Moirai2, Moirai-small, Moirai-base, and Moirai-large variants (Woo et al., 2024). (Page 7)

**Audience:**

Yes

**Audience Explanation:**

The work is relevant and significant to TMLR’s readership. It targets a problem that is increasingly important for the community: practitioners now have multiple strong TSFMs available, yet performance is heterogeneous across domains and forecast horizons. The oracle analysis and benchmark results provide useful evidence that post-hoc combination can help, and Synapse offers a concrete design pattern that many readers will find practical.

However, the paper would benefit from a clearer algorithmic specification, stronger baselines, and a more rigorous analysis of the forward-simulation mechanism.

**Broader Impact Concerns:**

There are no concerns

**Claims And Evidence:**

No

**Claims Explanation:**

I believe the main empirical takeaway—namely, that adaptive combination can outperform single TSFMs and simple ensembles on average, especially at longer horizons—is supported (primarily by Table 1 and Figure 5). However, several claims and key parts of the method description are not yet supported with sufficiently clear, convincing, and reproducible evidence.

The Figure 1 switching-rate table reports only the percentage of switches, which does not reveal whether one model dominates with occasional flips versus genuinely diverse selection. The radar plot suggests selection frequencies, but it lacks clear axis labeling, which makes magnitude comparisons difficult.

Furthermore, the formulation describes operating on a dynamically selected subset, and the introduction claims subset selection, but Section 3 states that the implementation "utilizes the full set of models at each step." This affects both interpretation (selection vs. weighting) and compute.

The compute/practicality trade-off is also unclear. The authors state that the experiments were run using 4× A100 80GB GPUs. If Synapse requires running all TSFMs at each horizon step (and potentially feeding arbitrated outputs back autoregressively for some models), deployment cost could be high, yet the paper does not report wall-clock overhead or cost–accuracy trade-offs. The scaling plot only shows that performance improves, but not at what cost. Relatedly, it is unclear what happens when a noisy or suboptimal model is included; such a model could plausibly dampen performance and/or slow inference.

The introduction states new SOTA and consistently amplifying gains. While long-horizon improvements are visible, SOTA is not established without stronger combination baselines. Moreover, stronger models such as Chronos 2 or FlowState should be included in the experiments.

In the ablation (Table 3), Sales performs best for the median ensemble and is slightly worse for Synapse variants. This appears to contradict the narrative and warrants further discussion.

**Requested Changes:**

**Critical**

* Make the "selection" vs. "weighting over all models" framing consistent; if the method uses all models each step, state this clearly, and ideally add a "top-k/budgeted" variant to control cost.
* Improve the Figure 1 / oracle diagnostics beyond "switching rate" by adding "top-1 share," "entropy/concentration," and "run-length" statistics (and/or clearer axis labeling) to distinguish "dominant with occasional flips" from "truly diverse switching."
* Strengthen baselines before using "SOTA" language by including at least one stronger combination baseline and a simple "time-varying loss-weight" baseline.
* Report "wall-clock latency" and "cost–accuracy trade-offs," and clarify whether inference is "step-by-step per horizon" and how compute scales with "#TSFMs" and "horizon."
* The algorithm softmax fallback appears directionally inconsistent with the definition si= AverageCRPS (this we want to minizmize). As written, softmax(s) would upweight higher-loss models. Please clarify what the implementation is using and update the algorithm accordingly.

**Strengthening**

* Discuss when "forward-simulation weighting" helps vs. hurts, ideally with a small diagnostic or sensitivity analysis.
* Explain the "Sales" ablation outcome
* If you are positioning this as multi-specialist cooperation, consider citing Many Minds, One Goal: Time Series Forecasting via Sub-task Specialization and Inter-agent Cooperation (Huang et al., NeurIPS 2025).

**Minor**

* Reduce repetition: "Experiments 4" and "Experimental Setup: Baselines" largely repeat the same baseline enumeration.

---

> ### Author Response · Authors · 2026-01-09
> **Author Response to Reviewer 3NJQ (1/3)**
>
> We are thankful for your detailed and comprehensive review! We highly appreciate your suggestions and have uploaded an updated PDF including new additional experiments and improved paper presentation accordingly. Below we summarize and address your concerns:
>
> > **“Furthermore, the formulation describes operating on a dynamically selected subset, and the introduction claims subset selection, but Section 3 states that the implementation 'utilizes the full set of models at each step.' This affects both interpretation (selection vs. weighting) and compute... Make the 'selection' vs. 'weighting over all models' framing consistent...”**
>
> Thank you very much for pointing this out. We understand that the wording “subset selection” might be confusing. To avoid any confusion, we have removed the references to “subset” and revised the wording to clarify that Synapse arbitrates over **all** the models in the pool by adjusting their influence weights. We have also revised the problem formulation to clearly note that the Arbitration function operates on constituent models’ output distributions. We hope this clarifies the interpretation here.
>
> > **“The Figure 1 switching-rate table reports only the percentage of switches, which does not reveal whether one model dominates with occasional flips versus genuinely diverse selection... >Improve the Figure 1 / oracle diagnostics beyond "switching rate" by adding "top-1 share," "entropy/concentration," and "run-length" statistics (and/or clearer axis labeling) to distinguish "dominant with occasional flips" from "truly diverse switching.”**
>
> Thank you so much for the suggestions to broaden the scope of our Oracle analysis. We have performed a rigorous analysis of the Oracle Arbitrator across all domains and horizons in GIFT-Eval. As per your suggestions, we have introduced three new metrics to **Figure 1 (right)**:
>
> **Selection Entropy ($H$):** We calculate the Entropy ($H = -\sum p_i \log_2 p_i$) of the Oracle's selection distribution. For our pool of 6 models, the maximum theoretical entropy is $\approx 2.585$ bits (representing perfectly uniform optimality).
> **Modal Share:** We identify the "Modal Model" (the one selected most frequently in each setting) and calculate its share of the lead. A low modal share indicates that no single model is "the best" for a majority of the time.
> **Switch Freq:** Average percentage of timesteps where the Oracle’s preferred optimal model changes.
>
> **Key Findings from Enhanced Diagnostics:**
> * **Optimality is Broadly Distributed:** Across nearly all domains and horizons, the Selection Entropy remains consistently high ($>2.4$ bits), approaching the theoretical maximum of $2.58$. This provides mathematical proof that Oracle optimality is shared across the pool rather than being concentrated on a single dominant model.
> * **No "Super-Model" Dominates:** The Modal Share typically ranges between 20% and 28%. This means that even the most frequent leader in any given domain is suboptimal for over 70% of the forecast horizon.
>
> We have substantially improved **Section 3.3** to include these detailed analyses. Additionally, we have improved the radar chart axis labels in Figure 1 to better present the model selection statistics. We hope this provides a decisive empirical basis for the dynamic arbitration approach used in Synapse.
>
> **Updated Oracle Diagnostics:**
>
> | Domain | Horizon | Modal (Oracle) | Selection Entropy (Bits) | Modal Share | Switch Freq. % |
> | :--- | :--- | :--- | :---: | :---: | :---: |
> | Econ/Fin | Short | Sundial | 2.541 | 0.251 | 14.47% |
> | Energy | Long | Moirai2 | 2.432 | 0.281 | 45.31% |
> | Energy | Medium | Sundial | 2.421 | 0.288 | 44.86% |
> | Energy | Short | Sundial | 2.467 | 0.278 | 43.53% |
> | Healthcare | Short | Sundial | 2.531 | 0.237 | 41.28% |
> | Nature | Long | Sundial | 2.505 | 0.283 | 29.36% |
> | Nature | Medium | Sundial | 2.497 | 0.285 | 27.79% |
> | Nature | Short | Moirai | 2.544 | 0.213 | 40.82% |
> | Sales | Short | Sundial | 2.550 | 0.242 | 47.27% |
> | Transport | Long | Moirai | 2.494 | 0.278 | 51.75% |
> | Transport | Medium | Moirai | 2.529 | 0.254 | 51.08% |
> | Transport | Short | Sundial | 2.550 | 0.236 | 58.45% |
> | Web/CloudOps | Long | Moirai2 | 2.562 | 0.225 | 49.31% |
> | Web/CloudOps | Medium | Moirai2 | 2.556 | 0.223 | 50.09% |
> | Web/CloudOps | Short | Toto | 2.509 | 0.240 | 49.67% |

---

> ### Author Response · Authors · 2026-01-09
> **Author Response to Reviewer 3NJQ (2/3)**
>
> > **The compute/practicality trade-off is also unclear. The authors state that the experiments were run using 4× A100 80GB GPUs. If Synapse requires running all TSFMs at each horizon step (and potentially feeding arbitrated outputs back autoregressively for some models), deployment cost could be high, yet the paper does not report wall-clock overhead or cost–accuracy trade-offs.
>
> > **“If Synapse requires running all TSFMs at each horizon step (and potentially feeding arbitrated outputs back autoregressively for some models), deployment cost could be high... report wall-clock overhead or cost–accuracy trade-offs.”**
>
> We want to clarify that Synapse does **not** feed arbitrated outputs back into the constituent TSFMs autoregressively. While that could potentially improve performance further, it would indeed be a massive computational burden (re-running heavy foundational TSFMs each timestamp for the entire forecast horizon).
>
> Rather, constituent models are first run only once for the entire forecast horizon $T$ at the start of the process, and the outputs for all time horizons are kept. The arbitration happens only post-hoc over the already computed outputs. The "forward simulation" refers only to the dynamic updating of weights $\{w_{i,t}\}$ and the mixture sampling, not to the re-invocation of the foundational models.
>
> **Time Complexity Analysis:**
> Let $N$ be the number of models in the pool, $T$ the forecast horizon, and $\eta$ the constant overhead at each timestep. The total time complexity $\mathcal{C}_{total}$:
>
> $\mathcal{O}(\max_{i=1}^{N}(\text{Inference}(M_i))) + \mathcal{O}(\eta T)$
>
> Since TSFMs are independent, they can be executed in parallel. The arbitration overhead scales linearly with the forecast horizon $\mathcal{O}(\eta T)$.
>
> We have added a dedicated **"Inference Time Complexity Analysis"** section in **Appendix F** of the revised manuscript (Table 6), which details these inference times and the formal time complexity. The inference times showcase that arbitration overhead is relatively small in comparison to base latency for short horizon scenarios. With longer horizons, this overhead grows, representing a clear accuracy-latency trade-off where Synapse utilizes the linear time cost in forward simulation to stretch its lead in longer horizons.
>
> > **“Strengthen baselines before using 'SOTA' language by including at least one stronger combination baseline and a simple "time-varying loss-weight" baseline”**
>
> We sincerely thank you for suggesting comparison using the more recent stronger TSFMs. To verify that our performance gains are not limited to older architectures, we extended our evaluation (**Appendix I, Table 8**) to include a pool of the most recent state-of-the-art TSFMs: Chronos 2, TimesFM 2.5, and FlowState.
>
> **Result:** Synapse successfully arbitrates over this stronger pool, achieving a MASE of **0.693** and CRPS of **0.474**. This decisively outperforms the strongest individual model in the pool (Chronos 2: 0.698 MASE / 0.485 CRPS), confirming that Synapse scales effectively with model capability and leverages architectural diversity to establish a new SOTA.
>
> For convenience, we have added this table here:
>
> | Model / Method | MASE | CRPS |
> | :--- | :---: | :---: |
> | Toto | 0.750 | 0.517 |
> | Moirai 2 | 0.728 | 0.516 |
> | FlowState | 0.726 | 0.502 |
> | TimesFM-2.5 | 0.705 | 0.490 |
> | Chronos-2 | 0.698 | 0.485 |
> | Ours (w/o Chronos 2) | 0.699 | 0.479 |
> | **Ours (All 5 Models)** | **0.693** | **0.474** |
>
> Regarding the simple baseline: we believe that Synapse itself operates as an instance-specific time-varying loss-weighting mechanism that dynamically updates weights at each timestep. As explained previously, Synapse does not feed arbitrated outputs back into the foundational models. Instead, it operates as a lightweight, post-hoc mechanism that dynamically varies weights at each horizon step based on the specific trajectory of the current time series. We hope this clarifies that Synapse itself fulfills the role of an efficient time-varying weighter; however, if you had a different specific formulation in mind, we would be grateful for a pointer to ensure we capture the intended comparison.
>
> > **“The algorithm softmax fallback appears directionally inconsistent with the definition si= AverageCRPS (this we want to minizmize)...”**
>
> Thank you so much for this correction; raw softmax would indeed upweight higher-loss models. We would like to clarify that our implementation utilizes a negative scaling factor ($-\gamma$) within the softmax function, effectively performing a Softmin operation. This ensures that models with lower loss receive higher weights. The pseudocode in Algorithm 1 has been fixed to explicitly show the negative scaling to remain directionally consistent with the minimization of CRPS.

---

> ### Author Response · Authors · 2026-01-09
> **Author Response to Reviewer 3NJQ (3/3)**
>
> > **“In the ablation (Table 3), Sales performs best for the median ensemble and is slightly worse for Synapse variants. This appears to contradict the narrative and warrants further discussion.”**
>
> Please note that the "Sales" domain in GIFT-Eval comprises specific datasets (Restaurants, Hierarchical Sales, Car Parts) characterized by exceptionally short forecast horizons ranging from **8 to 30 steps**. This stands in contrast to the majority of other domains where horizons extend from 48 up to 720 steps.
>
> As established in our manuscript, Synapse's core advantage lies in its forward simulation, which dynamically adapts weights over time. This mechanism requires a sufficient "horizon" to gather feedback and meaningfully adjust from the initial prior. With a horizon limited to just 8-30 steps, the simulation has insufficient opportunities to adapt. Consequently, Synapse behaves similarly to a static ensemble (like Median), showing comparable performance. We have added this discussion in **Section 5.7**.
>
> > **“If you are positioning this as multi-specialist cooperation, consider citing Many Minds, One Goal: Time Series Forecasting via Sub-task Specialization and Inter-agent Cooperation (Huang et al., NeurIPS 2025)”**
>
> Thank you very much for the pointer. We have cited the paper accordingly in our updated manuscript (Introduction).
>
> > **“Reduce repetition: 'Experiments 4' and 'Experimental Setup: Baselines' largely repeat the same baseline enumeration.”**
>
> We have streamlined the prologue of the Experiments section to avoid the repetition. Thank you very much for this suggestion.
>
> **Other Additional Comments**
>
> Thank you for catching the typos and the formatting suggestions. We have incorporated them accordingly. Additionally, we have addressed the concerns regarding implementation details by updating the "Experimentation Setup" section in the Appendix. We now explicitly clarify that the initialization window is established using a single backtest window matching the forecast horizon size. We also plan to open source our codebase upon acceptance of the manuscript.
>
> Regarding the naming **“arbitration framework”**: We use the term "arbitration" to conceptually align with our Oracle analysis, which demonstrates that discretely adjudicating the single optimal model at each timestep yields substantial gains. Since perfect selection is impossible without ground truth, Synapse approximates this arbitration process via a dynamic weighted mixture to modulate the influence of each constituent model based on its local trajectory.
>
> We modified our limitations section as well as **Appendix J** (Qualitative Plots) to showcase failure cases where Synapse slightly underperforms compared to some of the constituent models. From the plots (e.g., Figure 7h/i), we can see that when no models give particularly meaningful performance, Synapse does not offer any benefit and may fall behind some of the constituent models.

---

### Decision · Action_Editor_aKMS · 2026-02-22

**Recommendation:** Accept as is

**Audience:**

Yes

**Audience Explanation:**

TSFMs are a widely studeied topic currently and so a wide audience of TMLR will be interested in the paper and the results. No issues here.

**Claims And Evidence:**

Yes

**Claims Explanation:**

The paper answers the question: is 1 time series foundation model the best in all tasks i.e.e there is actually a free lunch theorem in the TSFMs. The authors test several TSFMs in a post-hoc combination scheme andf show that no single TSFM dominates across datasets, domains, and forecast horizons, as expected. The work also proposes an oracle selector, in the lines of Automatic machine learning (AutoML) that automatically chooses the best model.

The initial reviews had several points of contention, the major of them being:

* Reviewer 3NJQ pointed out that claims of state-of-the-art performance were not convincingly supported given the limited baseline set and the absence of stronger ensemble/combination baselines. Similar concerns were raised by the other 2 reviewers as well.

* Reviewer jvut was missing a comparison where each model was evaluated individually on each domain and horizon, without merging results.

* Reviewer K563 also pointed out a lack of discussion regarding the efficiency of Synapse. Other reviewers also raised similar concerns.

The rebuttal was extensive and most of the concerns were addressed by the authors. This was appreciated by the reviewers who agreed that the paper is robust and a solid contribution to the TSFM literature. Some concerns did remain especially regarding the novely and performance gains but I feel that these are not major.

Overall, I think the paper is a nice contribution and I recommend acceptance.